# Evolution of schooling drives changes in neuroanatomy and motion characteristics across predation contexts in guppies

Alberto Corral-Lopez [1,2,3,4] ✉, Alexander Kotrschal[2,5], Alexander Szorkovszky[6], Maddi Garate-Olaizola [2,4], James Herbert-Read [7,8], Wouter van der Bijl [1], Maksym Romenskyy[2,9], Hong-Li Zeng[10], Severine Denise Buechel[2,5], Ada Fontrodona-Eslava [2,11], Kristiaan Pelckmans[12], Judith E. Mank [1] & Niclas Kolm [2]

One of the most spectacular displays of social behavior is the synchronized movements that many animal groups perform to travel, forage and escape from predators. However, elucidating the neural mechanisms underlying the evolution of collective behaviors, as well as their fitness effects, remains challenging. Here, we study collective motion patterns with and without predation threat and predator inspection behavior in guppies experimentally selected for divergence in polarization, an important ecological driver of coordinated movement in fish. We find that groups from artificially selected lines remain more polarized than control groups in the presence of a threat. Neuroanatomical measurements of polarization-selected individuals indicate changes in brain regions previously suggested to be important regulators of perception, fear and attention, and motor response. Additional visual acuity and temporal resolution tests performed in polarization-selected and control individuals indicate that observed differences in predator inspection and schooling behavior should not be attributable to changes in visual perception, but rather are more likely the result of the more efficient relay of sensory input in the brain of polarization-selected fish. Our findings highlight that brain morphology may play a fundamental role in the evolution of coordinated movement and anti-predator behavior.

Animals regularly gather - for safety, for exploiting resources, or for mating. Group-living often leads to spectacular forms of collective behavior, and individuals in many taxa coordinate their movements in order to increase efficiency in foraging and traveling, or to confuse predators[1]. Collective motion has evolved multiple times in fishes and is widely understood as a behavioral adaptation aimed at reducing the risk of predation[2]. This behavioral adaptation is underpinned by the efficient acquisition of information from external cues through visual

[1]Department of Zoology and Biodiversity Research Centre, University of British Columbia, Vancouver, Canada. [2]Department of Zoology/Ethology, Stockholm University, Stockholm, Sweden. [3]Division of Biosciences, University College London, London, UK. [4]Department of Ecology and Genetics, Uppsala University, Uppsala, Sweden. [5]Behavioural Ecology, Wageningen University & Research, Wageningen, Netherlands. [6]RITMO Centre for Interdisciplinary Studies in Rhythm, Time and Motion, University of Oslo, Oslo, Norway. [7]Department of Zoology, University of Cambridge, Cambridge, UK. [8]Aquatic Ecology, Lund University, Lund, Sweden. [9]Department of Life Sciences, Imperial College London, London, UK. [10]School of Science, Nanjing University of Posts and Telecommunications, Nanjing, China. [11]Centre for Biological Diversity, School of Biology, University of St Andrews, St Andrews, UK. [12]Department of Physics and Astronomy, Uppsala University, Uppsala, Sweden. ✉e-mail: alberto.corral@ebc.uu.se

and lateral line sensory systems[3–6]. To date, we have a detailed understanding that coordinated movements in animal groups likely emerge from decision rules that individuals use to interact in groups (e.g. refs. 7, 8). Similarly, correlation-based analyses have revealed how predation levels are associated with variation in collective motion in wild populations (see for instance ref. 9). Yet, the causes of collective motion are still unclear, particularly how evolutionary changes in collective motion contribute to anti-predator specific situations, or what type of sensory information and information processing schooling fishes use to identify and avoid predators as groups.

The brain, as the central organ controlling locomotion, sensory systems and decision-making, can play an important role in the ability to coordinate movements and decisions made in the context of grouping. As such, variation in the anatomy of the brain could be an important mechanism behind the evolution of collective motion. The link between social factors and changes in multiple brain structures across taxa is well established[10–13], and well-studied in fishes where approximately half of marine and freshwater species come together in groups at different life stages[14]. For instance, a comparative study on Lake Tanganyikan cichlids showed that species with more complex social structures have larger telencephali and hypothalami[15]. These regions are part of the fish forebrain, which has an important function in social behavior[16], and has been associated with social competence in cleaner fish and social orienting in zebrafish (*Danio rerio*)[12,17]. In nine-spined stickleback (*Pungitius pungitius*), exposure to larger groups during development is correlated with larger size of the optic tectum, the visual center in the fish brain[18].

Despite its potential importance, empirical studies explicitly testing the association between neuroanatomy and collective motion are to date scarce. Brain regions associated with fish social behavior have also been identified in the few studies explicitly testing the link between neuroanatomy and schooling behavior. Lesion studies in goldfish (*Carassius auratus*) showed that individuals with ablated telencephalon exhibited reduced activity and association with conspecifics[19]. Also, a study on surface and cavefish populations of *Astyanax mexicanus* living in different light environments showed an underlying positive correlation between optic tectum size and schooling behavior differences between populations[20]. These few previous studies highlight the potential role of neuroanatomy in schooling, as well as the need to account for environmental variation in analyses.

Here we use artificial selection lines of guppies (*Poecilia reticulata*) with divergence in polarization, the degree of alignment of members of a group when swimming. Polarization-selected guppies offer a unique opportunity to empirically evaluate the link between evolution of general schooling behavior, specific anti-predator behavior and neuroanatomy[21]. In relation to many fish species that associate in large schools, guppies have relatively low schooling propensity, with high levels of variation across individuals and across populations as a function of external factors such as predation risk or food availability[22,23]. In our selection lines, intrinsic schooling propensity was increased in female guppies by over 15% compared to controls in just three generations by selecting on individuals that exhibited higher polarization, the level of alignment between individuals moving together in a group[21,24]. Previous assays in the polarization-selected females showed that differences in polarization were caused by the combined effect of the likelihood to align with neighbors' movements, the strength of alignment to larger groups and individual swimming speed[21].

Following directional selection for polarization, we used these lines to investigate collective motion characteristics with and without predation threat, as well as to study the association between increased schooling propensity and changes in brain anatomy and visual performance. For this, we performed a series of experiments and measurements to: (i) evaluate collective motion patterns and predator inspection of individuals from polarization-selected and control female guppies when swimming in groups, (ii) quantify brain region sizes with microcomputed tomography (microCT) of female guppies from polarization-selected and control lines, and (iii) perform comprehensive tests of visual capabilities of these fish, spanning eye size, visual acuity and temporal resolution measurements of individual fish. Through the study of collective behavior in an ecologically relevant setting, we identify potential evolutionary pathways for how selection for higher polarization is associated with brain structure size variation.

## Results

### Collective motion in response to predation threat in guppies following artificial selection

We investigated whether selection for higher polarization affected cohesiveness and how individuals from groups reacted in response to neighbor movement in a predation context. These behavioral decisions should have major fitness consequences in this species[25]. Specifically, we recorded and tracked fish in an experimental arena to obtain positional data and assessed collective motion of groups of eight guppies when exposed to an imminent threat, the presence of an artificial replica of a pike cichlid (*Crenicichla frenata*), a natural predator in wild populations of Trinidadian guppies. Furthermore, we exposed these groups to a non-predator-shaped object to allow for comparisons when presented to a novel object. These assays were performed in combination with open field tests (OFT's) on the same fish groups. In total, we obtained data for 83 polarization-selected groups and 81 control groups of approximately six months of age and of similar body size (polarization-selected: standard length mean [95% CIs] = 24.8 mm [24.0, 25.5]; control: standard length mean = 24.5 mm [23.8, 25.3]). Previous analyses of the data for OFT's in these groups provided evidence that selection for polarization altered individuals' speed, how individuals aligned with, and how individuals were attracted towards conspecifics during group motion[21].

Our analyses of collective motion of female groups exposed to a predator model and a novel object showed predictable results in relation to previous findings observed in OFT's. In the presence of a novel object or a predator model, we observed overall declines of 12% and 17% in the polarization of the groups, while individuals' speed showed 20-fold and 24-fold decreases in the presence of a novel object and a predator model respectively (Supplementary Table 1 and Fig. 1a). Previously observed differences between polarization-selected and control groups in these traits were still present in the presence of these stimuli in the experimental arena with approximate differences of 7% (Supplementary Table 2 and Fig. 1a). Overall, nearest neighbor distance towards conspecifics in female showed a 10-fold increase in tests with a novel object and with a predator model in relation to OFT's (Supplementary Table 1 and Fig. 1a). Such strong reduction in median nearest neighbor distance observed between fish in the presence of these stimuli likely lead to minimum possible values of interindividual distance in our experimental setup. Indeed, differences between polarization-selected and control lines in nearest neighbor distance observed in OFT's[21] were not present in the presence of a predator model and a novel object (Fig. 1a; Supplementary Table 2). For median speed and group polarization, the predator model elicited a stronger response than the novel object. Specifically, collective motion data showed that guppy groups showed a 4-fold reduction in speed and were 4.5% less aligned in the presence of a predator model (LMM$_{speed}$: Estimate$_{novel\ object\ vs\ predator\ model}$ = 3.910 [0.709–7.100], $t$ = 4.71; df = 338; $p < 0.001$; LMM$_{polarization}$: Estimate$_{novel\ object\ vs\ predator\ model}$ = 0.045 [0.022–0.067], $t$ = 2.87; df = 338; $p = 0.011$: Supplementary Table 1, Fig. 1a).

### Information processing in response to predation threat

**Group polarization spatial patterns.** We studied spatial patterns associated with collective motion in our experimental setup by

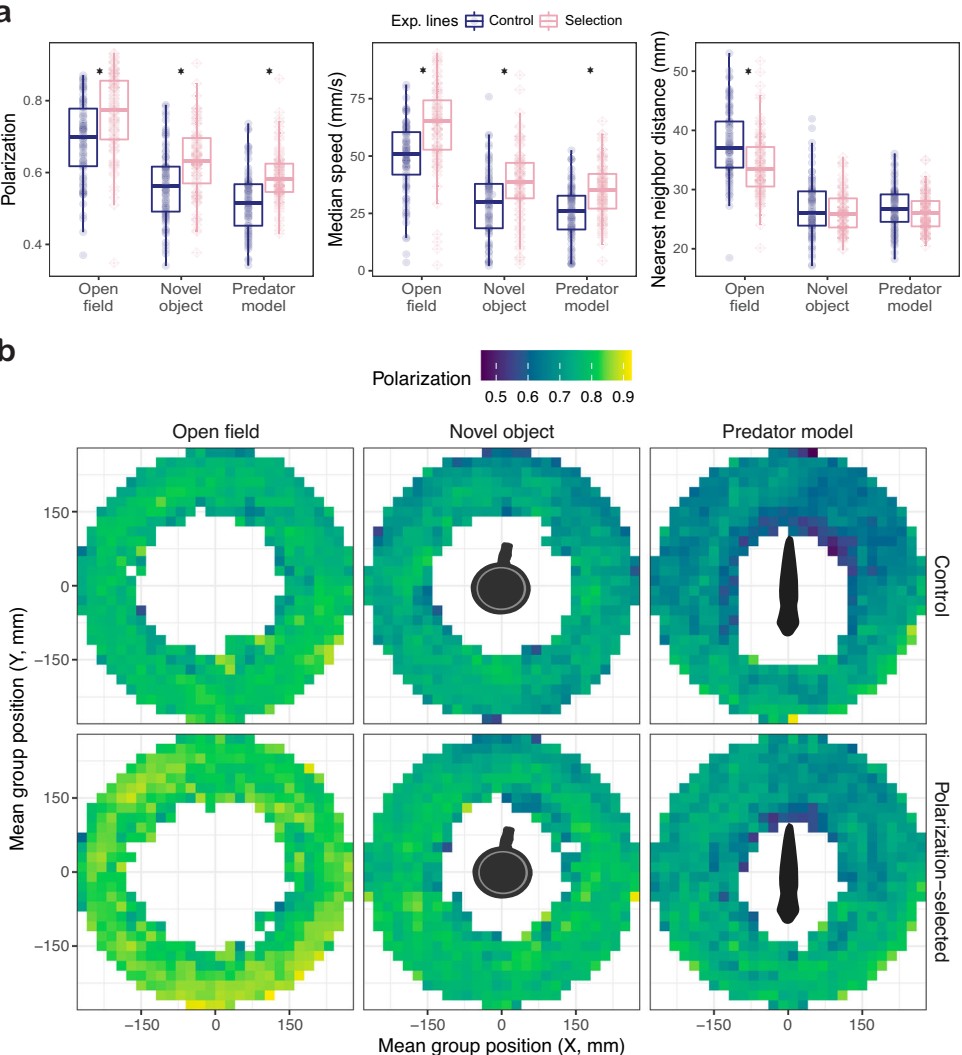

**Fig. 1 | Collective motion patterns in female guppies artificially selected for higher polarization. a** Boxplots of median polarization, speed and nearest neighbor distance for groups of eight individuals of polarization-selected (pink diamonds) and control (blue circles) female guppies assayed in an open field test ($n_{Polarization} = 89$, $n_{Control} = 85$), with a novel object ($n_{Polarization} = 86$, $n_{Control} = 84$) and with a predator model ($n_{Polarization} = 87$, $n_{Control} = 82$). Horizontal lines indicate medians, boxes indicate the interquartile range, and whiskers indicate all points within 1.5 times the interquartile range. Asterisks indicate $p < 0.05$ (see methods; Supplementary Tables 1 and 2). **b** Heatmaps of group polarization across different locations of the experimental arena when control (top) and polarization-selected (bottom) groups were exposed to open field, novel object and predator model assays. Grid cells with data for less than 8 groups were not depicted. Source data are provided as a Source Data file.

generating heatmaps with average values of group polarization and group centroid within a 20 × 20 mm grid for OFT's, predator model and novel object tests performed to polarization-selected and control groups. Visual inspection of heatmaps (Fig. 1b) concords with results obtained from statistical models using summary statistics for each group, suggesting a decline in polarization of the group in the presence of these stimuli. Yet, differences in polarization between control and polarization-selected groups in open field and novel object assays were consistent across all regions of the arena, while the observed differences in assays with a predator model were more pronounced in positions further away from the head of the model (Fig. 1b).

**Correlation between group polarization and speed.** To further characterize potential differences in how information spreads in polarization-selected and control fish groups we assessed the effect of group speed in the alignment of fish when exposed to a predation threat. For this, we generated heatmaps showing the correlation of group speed and group polarization across treatments for our

different selection lines (Fig. 2). Heatmaps indicated that group polarization of polarization-selected fish in OFTs was constantly close to maximal values of one and differences with control groups were partially independent of the speed of the group (despite larger values of median speed observed in relation to those in control groups). Overall group polarization reduction in the presence of a novel object and a predator model in relation to OFTs were likely associated with a strong reduction in the speed of the group, including periods of no movement of the group. Periods of no movement in response to a threat were observed more frequently in control groups suggesting that they might be an important driver of group polarization differences observed between polarization-selected and control groups. In addition to differences driven by lack of movement, correlations of group polarization and speed indicated that differences between polarization-selected and control groups observed during collective motion resembling motion in OFT's (mean group speed range 2–5 mm/frame) were maintained in the presence of stimuli in the experimental arena (Fig. 2).

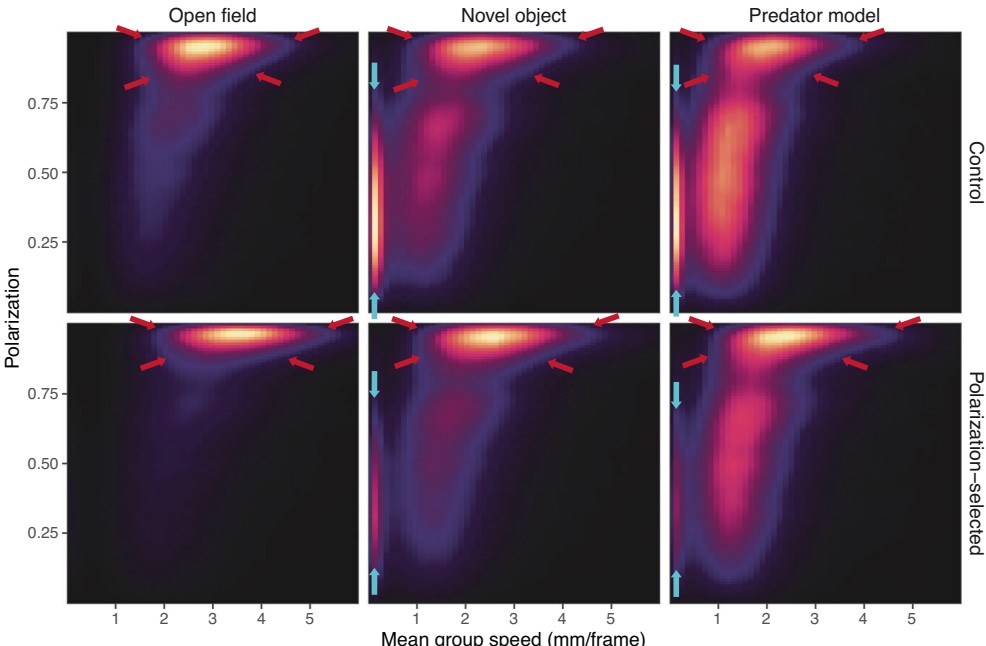

**Fig. 2 | Correlation between group polarization and speed.** Heatmaps showing the association between group polarization and speed in control (top) and polarization-selected (bottom) groups of eight guppy females exposed to open field, novel object and predator model assays. Group speed depicts the mean median swimming speed of the individuals in a group in mm per frame. Heatmaps indicate that the overall decrease in polarization in the presence of a novel object and a predator model was associated with a reduction in speed (higher relative density of group speed <2 mm/frame). The consistent differences in polarization across treatments between polarization-selected and control groups seem to be driven by: (i) lower reduction in mean group speed in polarization-selected groups, including periods of no movement (area delimited by blue arrows): and (ii) consistent differences in motion characteristics regardless of the presence of stimuli in the assay (lower range of group polarization at group speed >2 mm/frame, area delimited by red arrows). Source data are provided as a Source Data file.

**Predator inspection behavior.** Group polarization spatial patterns and its relationship to swimming speed suggested differences between polarization-selected and control fish in their attention to stimuli presented in the experimental arena. As such, we quantified inspection behavior of individuals in our groups of eight fish in the presence of a predator model. We scored recorded videos for the start and end point for each predator inspection performed by one randomly selected fish in each video. Analyses of predator inspection data showed that females from control lines presented a higher tendency to inspect the predation threat presented in the experimental arena than polarization-selected females. Specifically, we observed that polarization-selected females spent 21% less time inspecting the predator model and the mean duration of predator inspections were 18% shorter in polarization-selected females (GLMM$_{time\_inspecting}$: Incidence Rate Ratio (IRR)$_{polarization}$ = 0.79 (0.66–0.95), $t$ = −2.52, $p$ = 0.011; GLMM$_{mean\_inspection}$: IRR$_{polarization}$ = 0.82 (0.66–0.96), $t$ = −2.49, $p$ = 0.013, Fig. 3, Supplementary Table 3): We observed a similar trend for total number of predator inspections performed with a 13% reduced frequency of inspections observed in polarization-selected females (GLMM$_{inspections}$: IRR$_{polarization}$ = 0.87 (0.75–1.01), $t$ = −1.85, $p$ = 0.064; Fig. 3a, Supplementary Table 3).

**Collective motion during predator inspections.** Analysis of positional data and median distance to the stimulus presented in our assays suggested that most inspection behaviors to the predator model were performed during the initial 3 min of the assays (Supplementary Fig. 1). Further, the majority of inspections were performed at a range <200 mm and in the tail area of the predator model presented in the experimental arena (Figs. 1a and 3d). Consequently, we filtered our data to evaluate collective motion patterns of fish

groups in the time and locations where predator inspections were performed during our assays (see methods). The overall differences in group polarization found between polarization-selected and control groups were maintained in areas within 200 mm of the predator model (Fig. 3e; LMM$_{polarization<200mm}$: Estimate$_{selection}$ = −0.044 [−0.088,−0.000], $t$ = −1.984, df = 272, $p$ = 0.048; Supplementary Table 4a, b). Inspection behavior is mainly performed from areas with reduced risk of attack from a predator. In line with such expectation, we found that polarization of all groups was greatly reduced in the area of the predator model tail (Fig. 3e). However, we found no differences in group polarization between selected and control females in the head area of the predator model, but a maintenance of 8% differences in group polarization between polarization-selected and control females in close proximity to the tail of the predator model (LMM$_{polarization – head area}$: Estimate$_{selection}$ = −0.037 [−0.102, 0.027], $t$ = −1.53, df = 4.58, $p$ = 0.190; LMM$_{polarization – tail area}$: Estimate$_{selection}$ = −0.080 [−0.146, −0.013], $t$ = −1.53, df = −3.48, $p$ = 0.030; Fig. 3e, Supplementary Table 5a, b).

### Changes in neuroanatomy following artificial selection
We used microcomputed tomography (microCT) to reconstruct the brain anatomy of 13 polarization-selected females, and 15 control females and determine overall brain volume and the volumes of 11 major brain regions that could be safely identified in these brain scans covering the whole brain volume: olfactory bulbs, ventral telencephalon, dorsal telencephalon, thalamus, hypothalamus, nucleus glomerulus, torus semicircularis, optic tectum cup, central optic tectum, cerebellum, and medulla oblongata (see methods). We next used this data set to evaluate potential associations between collective

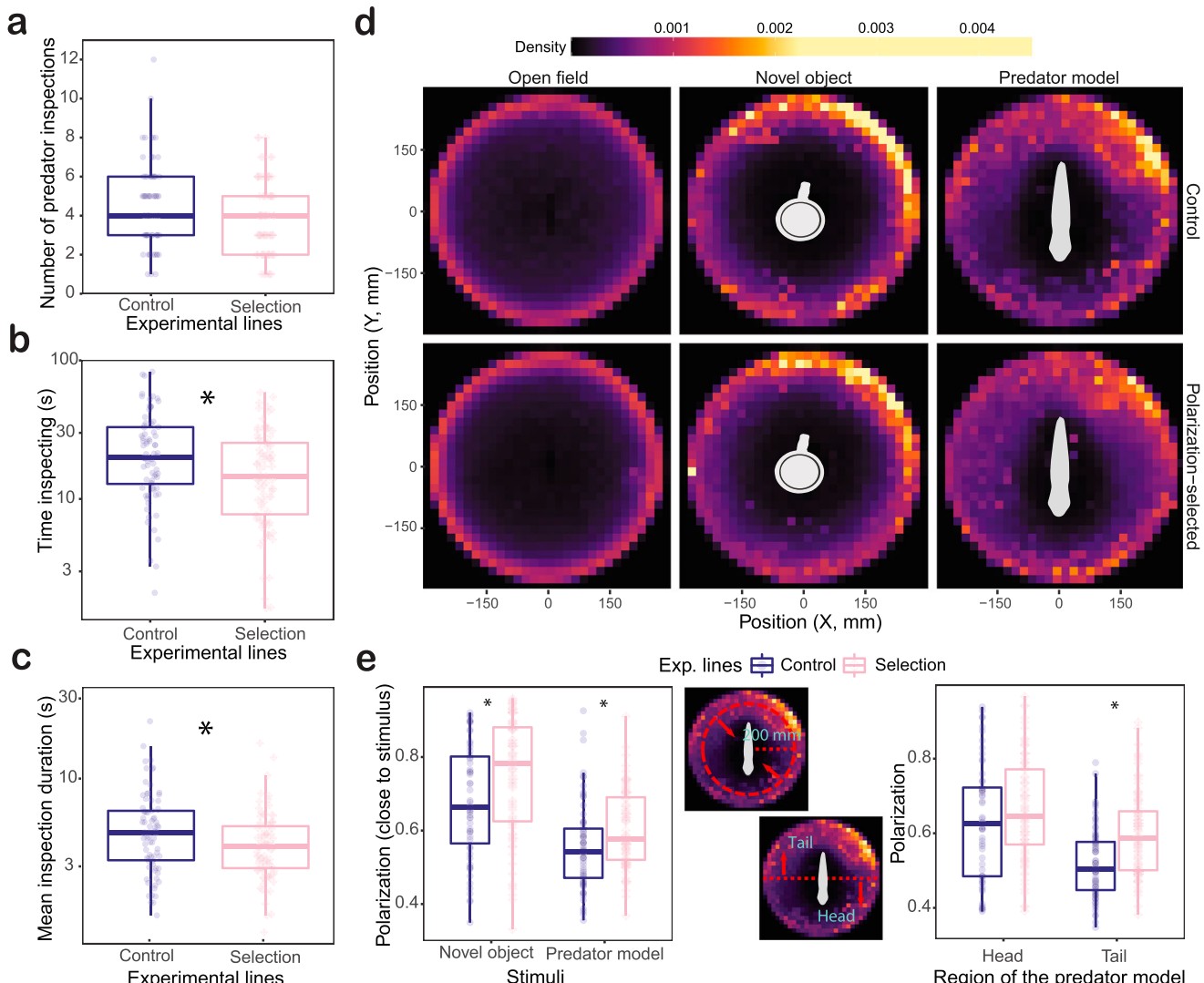

**Fig. 3 | Predator inspection behavior and collective motion.** Boxplots of number of predator inspections (**a**), total time inspecting (**b**), and mean inspection duration (**c**) for individuals when swimming in a group of eight polarization-selected (n = 86, pink diamomds) and control (n = 78, blue circles) female guppies in the presence of a predator model. **d** Density maps based on positional data of control (top) and polarization-selected (bottom) groups exposed to open field, novel object and predator model assays. **e** Boxplots of median group polarization in locations closer than 200 mm of the stimulus presented (left) and in the head and tail area of a predator model (right) in polarization-selected ($n_{Pol<200}$ = 82, $n_{HeadvsTail}$ = 86, pink diamonds) and control ($n_{Pol<200}$ = 70, $n_{HeadvsTail}$ = 75, blue circles) female guppy groups. For all boxplots in the figure, horizontal lines indicate medians, boxes indicate the interquartile range, and whiskers indicate all points within 1.5 times the interquartile range. Asterisks indicate $p < 0.05$ (see "Methods", Supplementary Tables 3–5). Source data are provided as a Source Data file.

behavior and neuroanatomy. Polarization-selected and control fish showed no differences in relative brain size (whole brain volume in relation to their body size; $LMM_{relativebrain}$: $t = -0.41$, df = 23.29, $p = 0.682$; Fig. 4a). However, analyses of relative brain region size (volume of each region in relation to the volume of the rest of the brain) indicated that the thalamus and optic tectum cups are approximately 7% and 4% larger in polarization-selected than control females respectively ($LMM_{thalamus}$: Odds Ratio (OR) $_{polarization}$ = 0.929 (0.866–0.998); $t = 2.187$, df = 25, $p = 0.038$; $LMM_{o.tectum}$: Odds Ratio (OR) $_{polarization}$ = 0.959 (0.924–0.994); $t = 2.409$, df = 23.09, $p = 0.024$; Fig. 4a), and the medulla oblongata is an 8% larger in control females ($LMM_{medulla}$: Odds Ratio (OR) $_{polarization}$ = 1.08 (1.01–1.14); $t = -2.65$, df = 23.91, $p = 0.013$; Fig. 4a). All other eight brain regions measured presented no differences between polarization-selected and control females in relative region volumes (Fig. 4a, b, Supplementary Table 6).

In parallel, we analyzed brain region volume differences using a more conservative approach and found similar and consistent differences between selection lines. Specifically, we used a multivariate

Bayesian model that included the relative size of the 11 brain regions as dependent variables. Posterior samples drawn from the multivariate model indicated that confidence intervals for the difference in relative volume in the medulla oblongata, the optic tectum cups and the thalamus between polarization-selected and control females did not overlap with zero (Supplementary Table 7a).

We then used the multivariate Bayesian model to evaluate the correlation in relative brain region volume between multiple regions measured. We focused on evaluating correlations with other brain regions for the three regions significantly differentiated between lines following artificial selection (Supplementary Table 7b). We found no correlation between optic tectum cup relative volume and volume of any other region measured. However, we found a significant inverse correlation between thalamus and medulla relative volume ($rescorr_{Medulla-Thalamus}$ [95% CIs]: −0.40 [−0.65/−0.12]). This finding suggests that the opposite differences observed in the volume of these two brain regions between control and polarization-selected female guppies may be linked to changes in brain

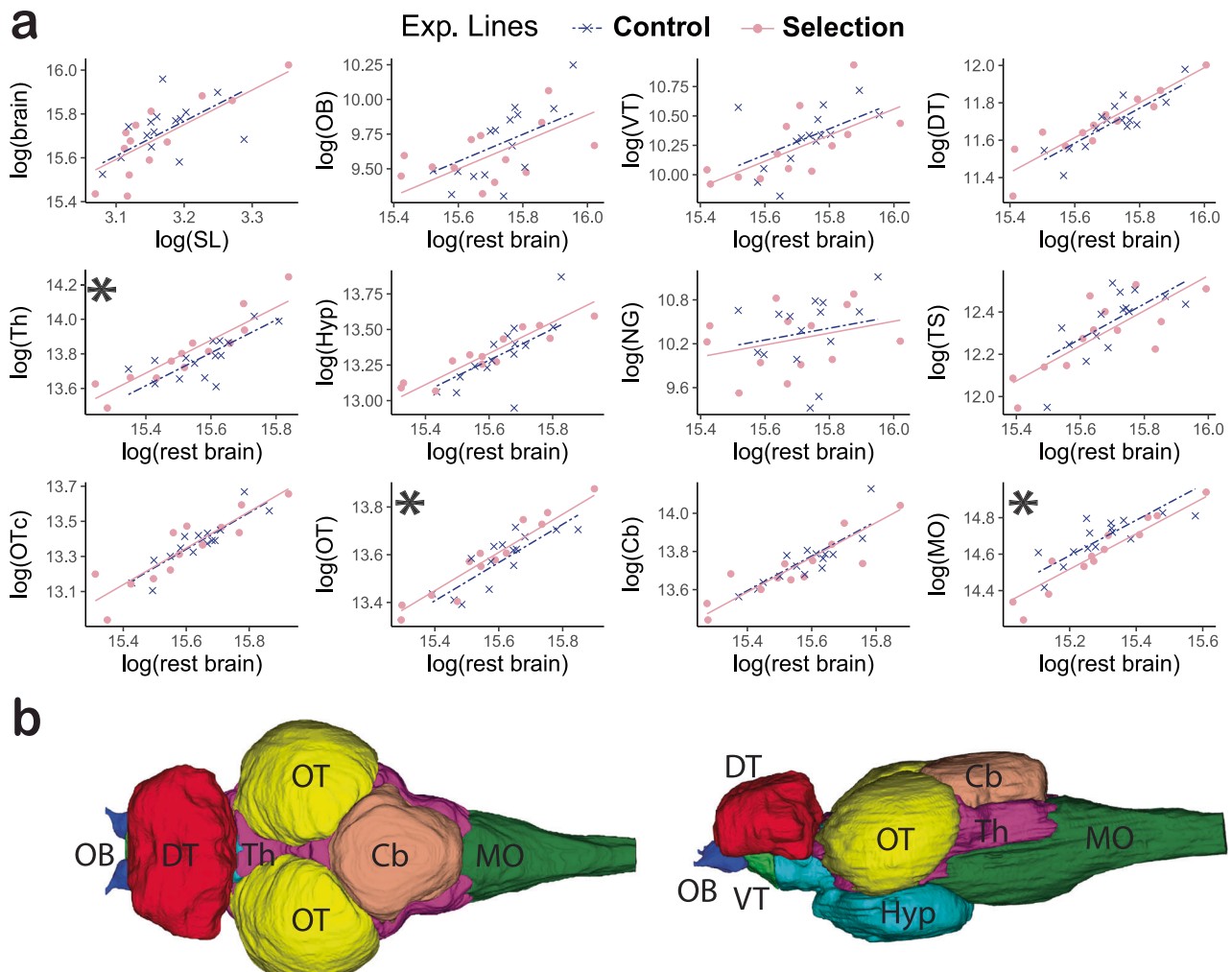

**Fig. 4 | The effect of artificial selection for higher polarization in neuroanatomical allometric relationships. a** The top left panel shows the allometric relationship between whole brain size volume and standard length of the fish (SL). Remaining panels show the relationship between each separate brain region with the rest of the brain ordered rostrally to caudally. Asterisks indicate brain regions with non-overlapping confidence intervals between polarization-selected females (pink circles; *n* = 13) and control females (blue crosses; *n* = 15) in two consistent statistical analyses (Supplementary Tables 6 and 7). **b** Reconstructed brain regions from microCT - scanned guppy brains. A dorsal (left) and lateral (right) view of a guppy brain with the major brain regions color coded: olfactory bulbs (OB; dark blue), dorsal telencephalon (DT; red), ventral telencephalon (VT; light green), optic tectum (OT; yellow), hypothalamus (Hyp; turquoise), thalamus (Th; purple), cerebellum (Cb; brown), medulla oblongata (MO; dark green). Not visible but measured brain regions include: torus semicircularis, nucleus glomerulus and optic tectum core. Source data are provided as a Source Data file.

development processes associated with artificial selection for higher coordinated motion.

## Information acquisition through the visual system

Efficiently acquiring information through the sensory system, mainly through visual cues, is a basic principle of collective motion in shoaling fishes[3]. Given observed differences in the size of the optic tectum cups between polarization-selected and control fish, we investigated potential differences in visual perception between lines. For this, we compared eye size and two key characteristics of the visual system to track movement of conspecifics, visual acuity and temporal resolution.

**Eye size.** We quantified eye size in females from polarization-selected (*n* = 57, standard length mean [95% CIs] = 27.0 mm [26.6, 27.4]) and control lines (*n* = 55, standard length mean [95% CIs] = 27.1 mm [26.8, 27.5]). Eye size is a common indicator of visual capacities of organisms[26], and comparative studies across fish species suggest that larger eyes correlate with improved visual abilities[27]. In our study, we found no difference between polarization-selected and control lines in

either absolute eye size or relative eye size, the proportional size of the eye in relation to body size (Fig. 5a; LMM$_{eye size}$: Estimate$_{selection}$ = −0.03 [−0.24, 0.19], *t* = −0.52, df = 2, *p* = 0.66; LMM$_{relative eye size}$: Estimate$_{selection}$ = −0.00 [−0.01, 0.01], *t* = −0.13, df = 2, *p* = 0.90; Supplementary Table 8). As previously reported in guppies[28], we found a strong positive correlation between eye size and body size (Pearson's correlation test: *t* = 11.72, df = 110, *p* < 0.001, cor [95% CIs] = 0.74 [0.65, 0.82]), but this correlation did not differ between polarization-selected and control lines (LMM$_{eye size}$: Estimate$_{selection*body size}$ = 0.010 [−0.016, 0.039], *t* = 0.75, df = 101, *p* = 0.45).

**Visual acuity.** We further assessed potential differences in visual perception between selection and control lines by quantifying visual acuity in the same individuals for which eye size was measured. Visual acuity allows an individual to resolve spatial detail and can be critical for an organism's fitness[29]. We measured visual acuity in our fish by quantifying their innate optomotor response in contrasting rotating gratings. This a widely used method to study visual acuity in multiple fish species, including guppies[30–32], and we have previously used this

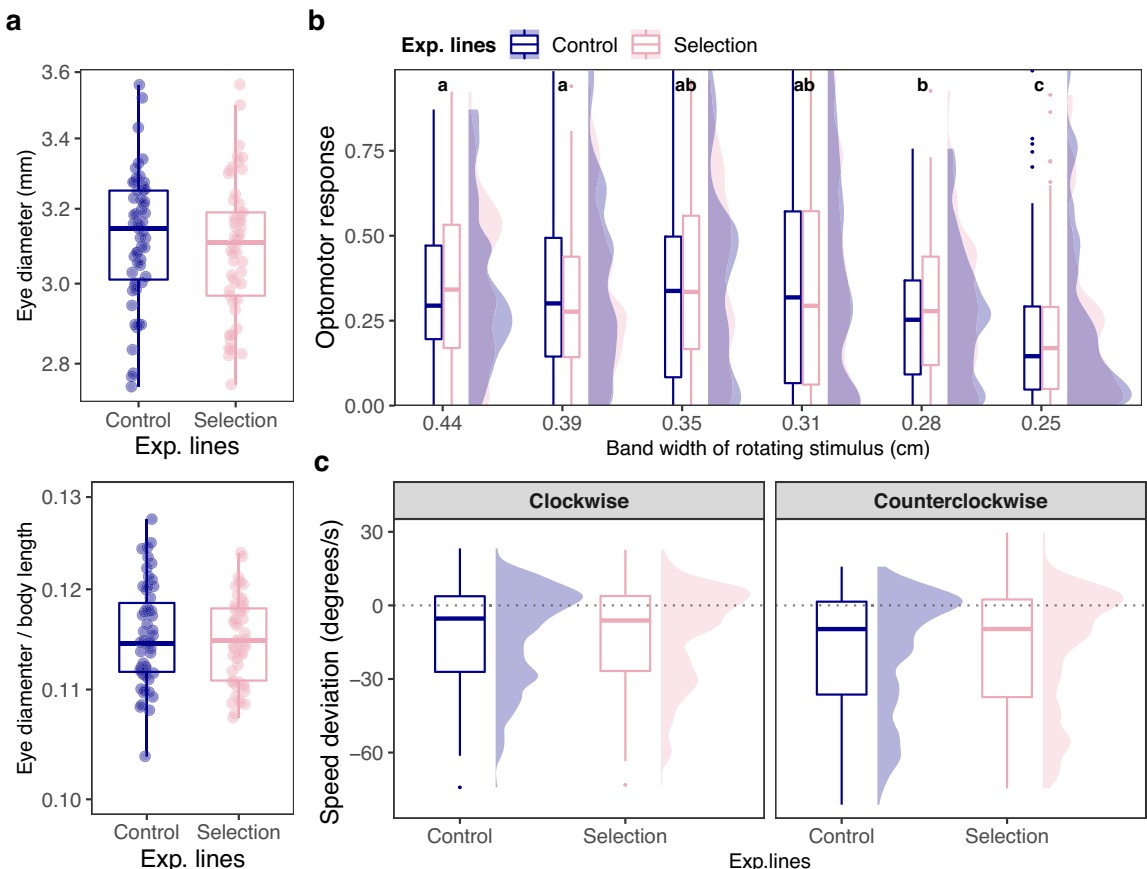

**Fig. 5 | Eye size and visual capacities of female guppies artificially selected for higher polarization. a** Boxplots of eye morphological measurements. **b** Boxplots and density plots of the proportion of time following 6 different rotating stimuli with rotating and static gratings of different widths at the lower end of guppy visual acuity (thinner widths represent a higher degree of difficulty to be perceived). **c** Boxplots and density plots of the deviation of fish swimming speed in relation to the speed that a rotating stimulus presented. in polarization-selected. For all morphological measurements and vision assays we measured the same polarization-selected ($n_{eyesize}$ = 57, $n_{optomotor}$ = 59, $n_{resolution}$ = 59, pink) and control females ($n_{eyesize}$ = 55, $n_{optomotor}$ = 57, $n_{resolution}$ = 55, blue). In all boxplots, horizontal lines indicate medians, boxes indicate the interquartile range, and whiskers indicate all points within 1.5 times the interquartile range. Optomotor response average values not sharing any letter are significantly different ($p < 0.05$) in post-hoc contrasts (see Supplementary Table 9b). No significant differences were observed for any comparison between control and polarization-selected fish (see Supplementary Tables 8–10). Source data are provided as a Source Data file.

approach to evaluate the visual system of guppies in similar contexts[28,33]. Following the methods in ref. 28, we exposed our fish to a series of six stimuli with rotating and static gratings of different widths at the lower end of the known guppy visual acuity, where thinner widths are more difficult to perceive. If polarization-selected and control groups differed in their ability to resolve spatial ability, differences in their optomotor response towards these rotating stimuli at the lower end of the guppy visual acuity should be expected. However, we found no difference in their average optomotor response combining data from all stimuli (LMM$_{acuity}$: Estimate$_{selection}$ = 0.00 [−0.08, 0.009], $t = 0.11$, df = 12.88, $p = 0.913$; Supplementary Table S9a), or in analyses independently evaluating specific optomotor response for any of the six stimuli presented (Fig. 5b; Supplementary Table 9c).

**Visual resolution tracking movement.** Although the ability to resolve spatial detail, acuity, is arguably an important visual parameter for guppies to recognize conspecific positions in shoals, it provides no information on an individual's ability to track movement[34]. Similar to many social fish species, guppies swim with a saltatory movement style that features discrete changes in speed and direction[9]. Consequently, we implemented an additional experiment that evaluated potential differences between polarization-selected and control fish in their temporal assessment of speed and direction changes. Using the same experimental apparatus used to evaluate visual acuity, we video

recorded female guppies from our selection and control lines and exposed them to a single-width rotating stimulus that moved in multiple directions and at different speeds (see "Methods"). We next used automated tracking to obtain orientation and speed of the fish for each frame and to quantify their direction and speed in relation to the stimuli presented at each time point. If polarization-selected and control groups differed in their ability to track movement, differences in the time following the correct direction of the stimuli and the deviation of their speed in relation to the stimulus speed should be expected.

Overall, fish followed the correct direction of the rotating stimulus for a significant proportion of the time. This was the case when the stimuli were presented in both a clockwise direction an in counterclockwise direction (Supplementary Fig. 2 and Supplementary Table 10). Overall, swimming speed did not significantly deviate from the stimuli rotating speed at the two lowest speed levels (Supplementary Fig. 2 and Supplementary Table 10), but was less than the stimuli speed at the two highest speed levels (Supplementary Fig. 2 and Supplementary Table 10). This was true for both directions in which stimuli were presented, but the mismatch between swimming and stimuli rotation speed was greater at the higher speed when the stimulus rotated anticlockwise (Supplementary Fig. 2 and Supplementary Table 10).

We compared the performance of polarization-selected and control fish in the test to evaluate their visual temporal resolution

while shoaling. We found no differences between polarization-selected and control lines in their deviation of their swimming speed in relation to the stimuli rotating speed for their combined scores across speeds and direction of rotation (Fig. 5c; LMM$_{speed\ deviation}$: Estimate$_{selection}$ = −0.46 [−8.74, 7.83], $t$ = −0.46, df = 2.64, p = 0.863; Supplementary Table 10a), or for their speed observed at any particular speed at direction of rotation (Supplementary Fig. 2 and Supplementary Table 10a). Similarly, polarization-selected and control females spent similar proportions of time following the stimuli during changes in stimuli rotating speed (Supplementary Fig. 2; Supplementary Table 10a; LMM$_{proportion\ time}$: Estimate$_{selection}$ = 0.00 [−0.14, 0.15], $t$ = 0.10, df = 2.52, $p$ = 0.928).

## Discussion

Our work demonstrates that selection for schooling behavior in female guppies has important implications for how this species behaves in the presence of potential threat. Analyses of motion patterns in these fish show that polarization-selected groups maintain higher polarization and speed when exposed to a potential threat. In addition, our analyses indicate that individuals from polarization-selected groups spend less time inspecting the threat than individuals from control lines. We further studied visual capacities in these fish and find no differences between polarization-selected and control individuals in visual acuity, temporal resolution or eye size, suggesting that effective processing of visual information in the brain is key for the observed differences in their ability to synchronize their swimming with close neighbors at multiple contexts.

In parallel, our results suggest that artificial selection for higher polarization has produced significant changes in the brain anatomy of female guppies. Neuroanatomical measurements indicate that polarization-selected fish exhibit a larger thalamus and a large optic tectum cup, but a smaller medulla oblongata, compared to control fish. These rapid changes in brain region sizes in response to selection for polarization behavior are consistent with previous artificial selection directly on neuroanatomy, which resulted in rapid shifts in both relative brain size and relative telencephalon size, in just a few generations in guppies[35,36].

Below, we discuss the implications of these discoveries for potential associations between brain morphology, anti-predator behavior and the evolution of collective behavior.

### Information processing in a predation threat context

Speed plays a prominent role in structuring fish schools, as individuals adjust their distance and alignment with neighbors through changes in speed[21,37,38]. Indeed, empirical studies across several species, as well as modeling approaches, have recently demonstrated a positive correlation between individual's speed and group polarization[38–43]. In our experiments, the presence of a novel object and a predator model prompted strong reductions in the median speed of individuals in the test. Yet, similar differences between polarization-selected and control females in individual's median speed and polarization were maintained across all contexts. Swimming performance tests previously performed in these fish demonstrated that selection did not target fish with better physiological capacities to swim faster, but those showing higher speed in response to neighbor positions[21]. Taken together, our results from these selection lines suggest that selection for coordinated movement is largely mediated by individuals' speed, and that group coordination characteristics such as polarization can emerge from individuals' tendencies to group with conspecifics and fitness benefits acquired from such tendencies at specific contexts.

Schooling in fishes is widely understood as a behavioral adaptation to escape the effect of predation[1]. These synchronized movements have been shown to confer two major benefits to schools, facilitating escape through transfer of information from closer neighbors[44], or by confusing the predator in which individual to attack[45]. The use of a static predator model in our assays does not allow us to infer any potential benefit of higher polarization on the confusion effect towards predators. However, our results show that directional selection and associated changes in the brain are correlated to behavioral changes in the presence of a threat and that it might affect individual ability to acquire private information about a potential predator. The reduced time spent inspecting the predator model by polarization-selected females, coupled with the fact that polarization-selected groups remained more polarized closer to the predator model, especially around the tail of the predator, suggest this a likely possibility. However, further comparisons within asocial and social contexts should be implemented to disregard the alternative explanation that directional selection leads to changes in predator inspection behavior also when fish have no access to social information.

Our study only measured fitness effects indirectly, using a predator model. Yet, previous work demonstrated that shorter inspection times towards the same predator models are associated with higher survival in the species[46,47]. Experiments with virtual prey analyses have also shown that individuals in groups with higher polarization are at lower risk of predation[45]. As such, our results following directional selection for polarization are consistent with the idea that selection acting on how individuals interact during motion might confer important fitness benefits for individuals showing higher coordination with conspecifics and reinforced in specific contexts, such as high-predation environments. Yet, such benefits might trade-off with a reduced level of private information obtained by individuals in relation to potential threats or foraging sources. In sticklebacks less sociable individuals are more likely to discover food first in a foraging context[37]. This agrees with our results in experiments with a predator model, where faster individuals' speed and higher tendencies to group with conspecifics seem to drive the observed lower time inspecting the predator model in the arena and acquire private information on a potential threat. Overall, these factors are arguably important selective pressures in natural populations where guppies from high-predation habitats swim with higher coordination and in larger groups[39,48]. Indeed, guppies from higher predation populations have been shown to rely less on private information for foraging resources, than those from lower predation populations[49]. Further studies evaluating fitness effects of relying in social versus private information across different predation pressures is paramount to understand how anti-predator behavior and collective motion drive evolutionary patterns at the proximate level.

### Brain morphology and collective behavior

We found two brain regions that were larger in relation to the rest of the brain in polarization-selected fish, the optic tectum cup and the thalamus. The optic tectum is the terminus of a vast majority of optic nerve fibers and axons of retinal cells[50] and as such is the primary vertebrate visual center. Despite wide variation in optic tectum size across teleost species, this region is known to control eyes and head movement in fish, as well as forming instant representations of the immediate surroundings to transform visual cues into behavioral responses[50,51]. This function is primarily achieved in superficial layers of the tectum[52], which corresponds to the optic tectum cup region used in our neuroanatomical parcellation of major brain regions in the guppy[53].

The evolved differences in optic tectum cup size between polarization-selected and control female guppies we found are concordant with phenotypic plasticity findings in nine-spined stickleback where it was found that individuals reared in groups developed larger optic tectum that those reared individually[18]. Differences in the ability to acquire sensory input have previously been associated with differences in schooling propensity[54]. In our experiment, rapid evolution of higher polarization did not lead to changes in visual performance or eye size. Together, these findings suggest that differences found

between polarization-selected and control lines in this section of the optic tectum should have an effect in their ability to control body orientation during complex social maneuvers such as predator evasion and swimming with low interindividual distances, but not in sensory information acquisition. This is consistent with current considerations of the tectum as more than a visuomotor relay, with implementation of intrinsic processing of visual information to extract critical features and send processed outputs to the posterior thalamus and then to further decision-making regions of the brain[51].

The primary receiving region of processed inputs from the tectum, the thalamus, was also found to be enlarged in polarization-selected females. To date, the level of homology between the mammalian and teleost thalamic regions is under strong debate[55,56]. In mammals, the thalamus plays critical roles in the modification, filtering and distribution of sensory and mechanosensory information into decision-making regions of the brain[56,57]. Recent findings in zebrafish suggest that these tasks are performed mainly by the preglomerular complex, whose developmental origin might be outside the diencephalic area that contains the thalamus in mammals[58]. The neuroanatomical atlas we created to analyze relative region size does not allow us to assess whether this particular region is part of the thalamus area which has been enlarged in polarization-selected females. However, our results of strong gene expression changes in the telencephalon of polarization-selected and control females in response to decision-making across social contexts[59], are in agreement with a higher efficiency in the relay of processed information towards the telencephalon, potentially facilitated by an enlarged thalamus. Studies across teleosts and mammals do agree on that the thalamus have an important role in the development of glutamatergic and GABAergic relay nuclei, with crucial roles in neurotransmission and inhibition. Interestingly, functional characterization of genomic and transcriptomic analyses performed in these selection lines identified a crucial role of glutamatergic synaptic function in the evolution of schooling[59]. Taken together, our results suggest that changes in the development of glutamatergic relay nuclei driven by a larger thalamus might affect the ability of individuals to synchronize their movements in a shoal. In parallel, and while not directly addressed in this study, the regulatory role of the thalamus in aggressiveness is concordant with common expectations of lower aggression levels in group-living species (reviewed in ref. 60). As such, further quantifications of the anatomical characteristics of the thalamus in relation to aggression levels within and across species might be a promising avenue for future research.

In contrast to the thalamus and the optic tectum cup, we found that the medulla oblongata was smaller in polarization-selected lines. The medulla oblongata is an important relay center of nervous signals between the spinal cord and ascendant brain regions with well-described functions in teleosts. First, it has an important role in basic motor function through processing of mechanosensory stimuli from hydrodynamic information[61]. Consistently found across fish species, the lateral line nerves terminate in the medial and caudal octavo lateralis nucleus in the medulla[62]. As the lateral line is crucial for schooling through cues that allow fishes to assess neighbor changes in speed and direction, the reduced relative size of the medulla in polarization-selected lines could be associated with potential differences in the ability to integrate and process information received through these nerves. Yet, our previous studies evaluating motor control capabilities found no difference between polarization-selected and control female guppies[21,63]. Further studies evaluating information processing of mechanosensory input in these selection lines is nonetheless paramount, with special focus on low light and high turbidity conditions.

Finally, studies in fishes indicate that the medulla has an important role for processing somatosensory signals, with special emphasis in auditory and gustatory signals[64,65]. Specifically, the vagal lobe, part of the medulla, is described to have a prominent role in processing taste. Consistent with our results, the vagal lobe of larval reef fishes is larger in solitary as compared to more social species[66]. While not tested in this study, it may be that the reduction in medulla observed in polarization-selected lines might be associated with important changes in the auditory system and the ability to perceive different tastes. In line with this reduction in the size of the medulla oblongata, our results show that three other brain regions have significant hypoallometric relationships with the medulla (see Supplementary Table 4b): the cerebellum (motor control center), the thalamus and the hypothalamus (hormonal regulation center). Gene expression of *angiopoietin*-1, a locus implicated in brain tissue development, showed contrasting expression levels between the medulla and the thalamic and hypothalamic regions[67]. Based on this, we hypothesize that selection for more coordinated motion leads to a trade-off between general sensory capabilities that are not important in coordinated movements and specific sensory capabilities required to coordinate movement with neighbors. In the future, it will be interesting to investigate the association between schooling propensity, brain anatomy and potential trade-offs between sensory and mechanosensory capacities.

## Final remarks
Our empirical approach with behavioral assays on artificial selection lines with divergence in polarization show that collective motion differences are consistent in the presence of a predator threat and that predator inspection behavior varies between the selection lines and the control lines. Moreover, we reveal differences in neuroanatomy that could provide a mechanistic explanation to the observed behavioral differences. Based on our discoveries, we propose that changes in behavior are intimately intertwined with matching changes in brain morphology during the evolution of collective behavior.

## Methods
### Ethics
All experiments were performed in accordance with ethical applications approved by the Stockholm Ethical Board (Dnr:C50/12, N173/13, and 223/15). These applications are consistent with the Institutional Animal Care and Use Committee guidelines.

### Artificial selection for higher group polarization
We evaluated the association between brain anatomy and collective motion in female guppies following artificial selection for higher polarization. Extensive detail on the selection procedure can be found in ref. 21. In short, groups of female guppies were tested in repeated open field tests and sorted in relation to the mean polarization of the group, the degree to which the individuals of a group move with higher alignment[21,24,68]. For three generations, females from groups with higher polarization were bred with males from those cohorts to generate three up-selected polarization lines. In parallel, random females were exposed to the same experimental conditions and bred with unselected males to generate three control lines. Third generation polarization-selected females presented on average a 15% higher polarization, 26% higher median speed and 10% higher group cohesiveness (i.e. 10% shorter nearest neighbor distances) in comparison to control females[21]. Further tests in these lines showed that selection for higher directional coordination was not only driven for selection for faster moving fish. This is because polarization-selected lines were still 5.7% more polarized after statistically controlling for speed differences, and the distance to conspecifics was an important factor driving differences in speed between polarization-selected and control lines[21]. The selection procedure targeted polarization on female groups and we found a weaker response to selection in males, and therefore subsequent neuroanatomical, behavioral and physiological studies focused on females. All fish were removed from their parental tanks after birth, separated by sex at the first onset of sexual maturation, and

afterwards transferred to single-sex groups of eight individuals in seven liter tanks. We kept all fish used for anti-predator response experiments, visual capacity tests and brain morphology measurements in these groups throughout their life span. Fish were not reused for experiments across these three categories. All tanks contained 2 cm of gravel with continuously aerated water, a biological filter, and plants for environmental enrichment. We allowed for visual contact between the tanks. The laboratory was maintained at 26 °C with a 12-h light:12-h dark schedule. Fish were fed a diet of flake food and freshly hatched brine shrimp daily.

### Anti-predator response in guppies following artificial selection
**Collective motion.** We evaluated anti-predator behavior in polarization-selected and control female guppies by conducting assays on 174 groups of eight fish in white arenas with 55 cm diameter and 3 cm water depth (polarization-selected: $n = 89$, standard length mean [95% CIs] = 24.8 mm [24.0, 25.5]); (control: $n = 84$, standard length mean [95% CIs] =24.5 mm [23.8, 25.3]). Each group was initially assessed in an open field assay in the arena for 10 min, and collective motion data from these open field assays was previously used to analyze differences in social interactions[21]. After 10 min, we sequentially introduced a novel object and a predator model for 6-min periods in the center of the experimental arena. In half the assays, we introduced the novel object first and the predator model second, with the order reversed in the other half of the assays. We used a blue coffee mug as a novel object and a fishing lure (18 × 3 cm) custom-painted to resemble the pike cichlid *C. frenata*, a natural predator on the guppy, as the predator model. These objects have been previously used to successfully reproduce natural behaviors of the guppy in response to a novel object and a predation threat[47]. To facilitate automated data collection, the position and orientation of the predator model was kept constant by using magnets in the bottom of the fishing lure and in the central position of the experimental arena. We provided external illumination to avoid shadowed areas in the circular arena. Disturbances were minimized by performing all assays in an isolated room of the laboratory. Under these conditions the presence of stimuli should be a major driver of the spatial locations of fish in the experimental arena. However, we are unable to quantify the effect of arbitrary spatial preferences in our results. Prior to the start of the assay, the eight-fish group was confined in the centre of the arena for 2 min in an opaque white 15 cm PVC cylinder. After this acclimation period, we lifted the cylinder and filmed the arena using a Point Grey Grasshopper 3 camera (FLIR Systems; resolution, 2048 pixels by 2048 pixels; frame rate, 25 Hz).

We tracked the movement of fish groups in the collected video recordings using idTracker[69] and used fine-grained tracking data to calculate speed, polarization and nearest neighbor distance in Matlab 2020 following methods established in ref. 70. For speed, we obtained the median speed across all group members by calculating the first derivatives of the x and y time series, then smoothed using a third-order Savitzky–Golay filter. For nearest neighbor distance, we obtained the median distance to the nearest neighbor for every fish across all frames. For group polarization, we calculated the median global alignment, which indicates the angular alignment of all fish in the arena. Calculations of median global alignment in each frame were only calculated if six out of the eight members of the group presented tracks following the optimization of our tracking protocol in the setup in ref. 24. No differences between completeness of tracks were observed between polarization-selected and control groups for any of the treatments in our experiment (Supplementary Fig. 3). For all measurements trials with less than 70% complete tracks were disregarded for further analyses. We additionally used R (v4.1.3), RStudio (v2022.07.0) and the tidyverse package[71–73] to generate heatmaps with average values of group polarization and group centroid within 20 × 20 mm grid cells for each frame (Fig. 1b). To avoid biases in group

centroid measurements potentially resulting from single individuals or small subgroups at large distances, we limited the analyses to frames in which at least six individuals formed a connected group, with an interindividual distance of 10 cm counting as a connection. Grid cells that did not contain values for a minimum of 8 groups per treatment were disregarded. No differences between the proportion of frames used were observed between polarization-selected and control groups for any of the treatments (Supplementary Fig. 3). To evenly compare motion patterns when presented with a novel object and a predator model to those obtained during the open field assays, we limited our analysis of the open field assay data to the initial 6 min of the recording.

We used LMMs with median speed, polarization and nearest neighbor distance as dependent variables to test for potential differences between polarization-selected and control lines. Selection regime, the type of stimulus presented and the interaction between these two were included as fixed effects, and body size of fish was coded as a covariate, with a random intercept for each replicated selection line and the order of presentation of stimuli as random factors. All models were run in R (v4.1.3) and RStudio (v2022.07.0)[71,73] using lme4 and lmerTest packages[74,75]. Model diagnostics showed that residual distributions were roughly normal with no evidence of heteroscedasticity. We obtained post-hoc comparisons of the response between selection line regimes at different levels of other fixed effects in the previous models using the emmeans package[76] with the tukey-adjustment method for multiple comparisons.

### Predator inspection behavior
**Behavioral scoring.** Positional data and analyses of median distance to center in our data indicated that groups of fish swam closer to the stimuli presented in the initial minutes following the addition of a predator model in the experimental arena, when compared to the same time periods following the addition of a novel object (Figs. 1a, 3d and Supplementary Fig. 1). This observation matched previous findings in similar experiments performed on guppies[47] and likely corresponds to the stereotypical behavioral response of guppies to inspect and gain information of a potential threat[25]. A predator inspection in guppies is characterized by an approach to the predator, monitoring predator activity and swimming sideways with an arched body. Based on this information, we manually visualized the videos during the first 3 min after addition of the predator model and scored the behavior of one randomly selected fish in the group using BORIS[77]. While blind to the selection line treatment, the start and end time of each predator inspection performed by the focal fish was scored for each video. We used the start and end time of predator inspections to calculate the number of inspections, average inspection duration and the total time that was spent inspecting per fish. Next, we fit a statistical model for each variable as a dependent variable using poisson or genpois distributions with log link function for the conditional mean in the package glmmTMb[78]. We used the selection line regime as a fixed effect. A random intercept for each replicated selection line, and the order of presentation of the stimuli in the arena were included as random factors in the model. We evaluated the adequacy of our fitted model using scaled-residuals quantile-quantile plots and residual versus predicted values plots in the DHARMa package[79].

**Group collective motion during predator inspections.** We analyzed positional data for each group by binning the observations in a grid, with 20 × 20 mm cells. For each trial, we calculated a density map, where the value for each grid cell was the fraction of all observations that occurred within that cell. The resulting density maps are a normalized representation on how often each grid cell was visited by individuals in our groups of eight fish when exposed to different stimuli (Fig. 2a). We used information from positional data to calculate summary statistics in different areas of interest. Predator model assays

presented unique spatial patterns in areas closer to the stimulus presented, with higher densities in the tail area of the predator model (Fig. 2a). Based on these factors, we calculated two new summary variables for each group: (i) median polarization of the group when the average position of the group was <200 mm to the predator model and novel object assays; and (ii) median polarization of the group in locations closer to the head (y-position > 0) and the tail (y-position < 0) in predator model assays. We used LMMs with these new calculated variables as dependent variables in the model to test for potential differences between polarization-selected and control lines. Selection regime, the type of stimulus presented or location in the tank were included as fixed effects, with a random intercept for each replicated selection line and the order of presentation of stimuli as random factors[74,75]. Model diagnostics showed that residual distributions were roughly normal with no evidence of heteroscedasticity. We obtained post-hoc comparisons of the response between selection line regimes at different levels of other fixed effects in the previous models using the emmeans package[76] with the tukey-adjustment method for multiple comparisons.

## Brain morphology of female guppies following artificial selection

We assessed neuroanatomical features of polarization-selected and control F3 females, approximately six months old (polarization-selected: $n = 15$, standard length mean [95% CIs] = 238 mm [235, 242]; control: $n = 15$, standard length mean [95% CIs] = 237 mm [232, 241]). We euthanized animals with an overdose of benzocaine and fixated the whole fish in 2% glutaraldehyde and 4% paraformaldehyde in phosphate buffered saline (PBS) for 5 days. Following two PBS washes, the brains were dissected out and stained for 48 h in 1% osmiumtetraoxide. We embedded the stained brains in 3% agar and scanned them using microcomputed tomography (microCT, Skyscan 1172, Bruker microCT, Kontich, Belgium). The scanner operated at a voltage of 80 kV, a current of 125 $\mu$m, with a 0.5 mm aluminum filter. Images were acquired using an isotropic pixel size of 2.4 $\mu$m. We reconstructed cross-sections from scanned images following a in NRecon (Bruker microCT) following a protocol successfully implemented in a previous study evaluating neuroanatomical differences between guppies up- and downselected for relative brain size[53]. This protocol allowed us to obtain measurements of whole brain size volume and brain region volume in 11 major brain regions in the guppy brain: olfactory bulbs, ventral telencephalon, dorsal telencephalon, thalamus, hypothalamus, nucleus glomerulus, torus semicircularis, optic tectum cup, central optic tectum, cerebellum, and medulla oblongata (Fig. 4b). We chose all regions that we could safely identify in the brain scans based on an adult swordtail brain atlas[80] and our own knowledge in fish neuroanatomy. Extended details on guppy brain region reconstruction from digital images can be found in ref. 53. Two brains from polarization-selected lines were damaged during the protocol, which reduced the sample size to 28 samples.

We tested for overall differences in relative brain size between polarization-selected and control lines using a linear mixed model (LMM) with brain volume as dependent variable, body size (standard length) as covariate, selection regime as fixed effect, and replicate as random effect. For each brain region, we used two different approaches to determine whether polarization-selected and control lines differ in relative brain region size. First, we used 11 independent LMMs with each region's volume as dependent variable, whole brain volume (excluding the volume of the region of interest) as covariate, selection regime as fixed effect, and replicate as random effect. LMMs were implemented in in R (v4.1.3) and RStudio (v2022.07.0)[71,73] using lme4 and lmerTest packages[74,75]. Second, to take into consideration that brain region volumes may be interdependent, we used a more conservative approach and analyzed the data using a Bayesian multilevel model that included 11 brain regions as dependent variables in a fully

multivariate context. The full model included an analogous structure to those used in the independent LMMs for each brain region. Parameter values were estimated using the brms interface[81,82] to the probabilistic programming language Stan[83]. We used default prior distributions with student-t distribution (3, 0, 2.5) for all parameters. The model estimated residual correlations among all brain region volumes with a Lewandowski-Kurowicka-Joe (LKJ) prior with $\eta = 1$, which is uniform over the range −1 to 1. Posterior distributions were obtained using Stan's no-U-turn HMC with six independent Markov chains of 4000 iterations, discarding the first 2000 iterations per chain as warm-up and resulting in 12,000 posterior samples overall. Convergence of the six chains and sufficient sampling of posterior distributions were confirmed by a scale reduction factor below 1.01, and an effective size of at least 10% of the number of iterations. For each model, posterior samples were summarized on the basis of the Bayesian point estimate (median), SE (median absolute deviation), and posterior uncertainty intervals by HDIs.

## Visual information processing in response to predation threat

**Visual acuity.** We evaluated the ability to perceive detail (visual acuity) in 9–12 months old female guppies (polarization-selected: $n = 59$, standard length [95% CIs] = 270 mm [266, 274], control: $n = 57$, standard length [95% CIs] = 271 mm [268, 275]). For this, we assessed their optomotor response, an innate orient behavior induced by whole-field visual stimulation[84], a widely used method to assess visual acuity across tax (reviewed in ref. 85). Extended methods and the optimization procedure for the stimuli used here can be found in ref. 28. Briefly, we projected a video recording with rotating vertical black and white bands of six different widths (stimuli) on the walls of a white ring-shaped arena of 25/50 cm of inner/outer diameter. Previous optimization of the methods found that the use of these stimuli allowed us to evaluate the optomotor response at the lower end of the species' acuity[28]. We placed individual fish in between the inner wall of the arena and a transparent ring of 40 cm diameter. After a 2-min acclimation period, we recorded their response towards six different rotating stimuli and the static images of these stimuli using a Sony Cam HDR-DR11E recorder. Each stimulus was presented for 1 min in random order. We manually scored the videos, recording the time that fish spent circling in the direction of the stimuli (clockwise) at a constant moving pattern using BORIS[77]. Behavioral scoring was performed blind to the treatment since only running numbers identified recordings. Likewise, scoring was blind to the rotation and bandwidth of the stripes since only the fish, but not the rotating stimuli, were visible during scoring. From the scoring, we calculated the proportion of time that a fish spent swimming in the direction of rotation of the stimuli, out of the total time that the different vertical black and white bands were presented to them.

**Temporal resolution.** Two weeks after visual acuity tests were completed in all fish, we measured the ability to track movement stimuli of different speeds (temporal resolution) in the same females from the polarization-selected ($n = 59$) and control lines ($n = 55$). We did not keep track of fish identity as fish were kept in groups with conspecifics of the same selection line and replicate between experiments. To evaluate temporal resolution, we placed fish in a white arena (50 cm diameter, 4 cm water depth) and exposed them to a projection of black and white bands of 3.5 cm width on the walls rotating clockwise and counterclockwise at four different angular speeds (14.4, 25, 36 and 45 degrees/s). We chose this bandwidth to focus on the assessment of fish response towards direction and speed changes, as this stimulus showed a maximal optomotor response in prior visual acuity tests[28]. The movement of each individual was recorded for a total of 1380 s with a Sony Cam HDR-DR11E; a 300 s acclimation period and 1080 s of clockwise and counterclockwise rotation of vertical bands at multiple speeds. Specifically, during each individual test, the 8 stimulus

combinations (4 speeds, 2 directions) were presented separately five independent times for 23 s. The total time of each individual test was 920 s (23 s per stimulus × 5 times during the test × 8 stimulus combinations). We randomized the order of presentation of different stimuli a priori, but this order was consistent for all fish. The stimulus changed speed with smooth transitions of 3 s, accelerating or decelerating to the next speed.

To quantify speed and direction changes of fish in our experimental setup, we automated behavioral scoring and obtained positional data using the Loopy Deep Learning Module (Loopbio 2020) in MATLAB (v. 2020a). X and y coordinates were transformed into a polar coordinate system centered on 0 and estimated from positional data. We calculated fish orientation by taking the difference in the fish's position between frames and defined their relative orientation (with respect to the arena) with the arcsin ($\sin(\theta - \vartheta)$), where $\theta$ was the orientation of the fish and $\vartheta$ is the angle of the arena radius going through the fish position. Positive values represent a fish swimming clockwise around the arena, while negative values represent swimming counterclockwise. For each frame, we identified whether the fish was swimming in the same direction of the stimulus projected and calculated the total proportion of time swimming in the direction of the stimulus. We also calculated the speed (in degrees per second) of the fish at each frame by using the dot product of the positional vector between consecutive frames. Using these values, we calculated for each individual the average total speed for each of the stimuli presented, and the average speed deviation between the speed of the stimulus presented and the speed of the fish.

**Eye size.** After the temporal resolution experiments were completed, we measured eye size in the females from polarization-selected lines ($n = 57$) and control lines ($n = 55$) that were previously assessed for visual acuity and temporal resolution. For morphological measurements, we anesthetized fish with 0.2 mg/l of benzocaine and took pictures of their left side. We measured eye diameter and body length in these pictures using ImageJ[86]. Relative eye size was calculated as the ratio of these two variables. Image analyses were performed by a single scorer who was blind to the selection line treatment in the photographs.

**Statistics.** Analyses were conducted in in R (v4.1.3) and RStudio (v2022.07.0)[71,73]. We analyzed potential differences in optomotor response, temporal resolution and eye size between polarization-selected and control females using LMM's. For the visual acuity trials, the proportion of time rotating was the dependent variable of the model. Fixed effects included selection line regime and bandwidth of the rotating stimuli. To account for differences in activity between fish, we used the proportion of time moving when presented with a static image of the stimuli as a covariate in the model. A random intercept for each replicated selection line, identity of the fish, and an observation-level variable were included as random factors in the model. For temporal resolution, we used selection line regime, the speed of rotation and the direction of rotation as fixed effects. The full model included the interaction between the selection regime with both speed and direction of rotation. This model included the identity of the fish, and a random intercept for each replicated selection line as random factors.

For eye size, eye diameter and relative eye size were dependent variables and models included selection regime as a fixed effect and a random intercept for each replicated selection line as a random factor. Model diagnostics showed that residual distributions were roughly normal with no signs of heteroscedasticity in optomotor response and eye size analyses. Model diagnostics on both models for temporal resolution analyses indicated unequal residual variance across the range of predicted values and a potential unequal influence of outliers. While estimates in linear mixed-effects models (LMMs) are argued to

be robust to violations of such assumptions[87], we used the robustlmm package to compare the estimates obtained with LMM's to robust models with the same predictors that provide reduced weights to outliers in the data[88]. Results were consistent regarding the modeling approach (Supplementary Table 11). We obtained post-hoc comparisons of the response between selection line regimes at different levels of other fixed effects in the previous models using the emmeans package[76] with the tukey-adjustment method for multiple comparisons.

## Reporting summary
Further information on research design is available in the Nature Portfolio Reporting Summary linked to this article.

## Data availability
Data needed to evaluate the conclusions in the paper are deposited in figshare database with https://doi.org/10.6084/m9.figshare.24080994[89]. Additional data related to this paper (video recordings) will be provided upon request from the authors. Source data are provided with this paper.

## Code availability
Code needed to evaluate the conclusions in the paper are deposited in figshare database with https://doi.org/10.6084/m9.figshare.24080994[89].

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

## Acknowledgements

We thank D. Sumpter for important contributions to the conceptualization of the artificial selection procedure. We thank A. Rennie, E. Trejo, M. Amcoff and A. Boussard for help with fish husbandry. This work was supported by the Knut and Alice Wallenberg Foundation (102 2013.0072 to N.K., and K.P.), the Canada 150 Research Chair Program, and the European Research Council (680951 to J.E.M.), the Swedish Research Council (2016-03435 to N.K., 2017-04957 to A.K., and 2018-04076 to J.H-R.), and the Whitten Lectureship in Marine Biology, University of Cambridge (to J.H-R.).

## Author contributions

N.K., K.P and J.E.M. contributed to conceptualization and funding acquisition of the project. A.K., A.S., M.R., S.D.B., J.H-R., H.Z., K.P., and N.K. contributed to the design of the selection procedure. A.C-L., A.K., A.S., A.F-E., M.R., S.D.B. conducted research to obtain collective behavior data and predator inspection data. A.K. H.Z. and K.P. conducted research to obtain neuroanatomical data. A.C-L., M.G-O., and J.H-R. conducted research to obtain visual capacities data. A.C-L. and W.vdB. performed formal analyses and visualization. A.C-L., J.E.M, and N.K. wrote the original draft. All authors contributed to the final version of the manuscript.

## Funding

## Competing interests

The authors declare no competing interests.
