## [Peer Review File · Nature Communications]

Evolution of schooling drives changes in neuroanatomy and motion characteristics across predation contexts in guppiesReviewers' Comments:

Reviewer #1:

Remarks to the Author:

I've read through the MS, "Evolution of schooling propensity in the guppy drives changes in anti-predator behavior that are linked to neuroanatomy," by Corral-Lopez et al, which explored links between neuroanatomy and coordinated movement in guppies. The authors found differences in the superficial layers of the optic tectum, thalamus, and medulla (but not characters related to visual perception) in artificial selection lines of female guppies, selected for divergence in polarization behavior.

Overall, I find this to be an excellent, well written MS that makes important steps forward in the field of comparative neuroanatomy. I find the questions that can be explored with artificial selection lines to be very interesting in terms of closing the gap between neuroanatomy and variation in behavior. My suggestions for improvement are relatively minor, but should be addressed prior to publication:

Introduction

-Pg 2, line 57: Note the plural of fish when discussing multiple species is fishes; a few instances of this throughout.

-I was surprised to see an almost exclusive focus on vision in relation to coordinated movement, with very little mention of mechanosensory cues (until the very last paragraph of the Discussion). While vision has definitely been shown to play a role in coordinated motion, it seems a bit of a glaring gap to neglect mention of hydrodynamic cues in schooling behavior until the end. I don't think it's a strong critique that the authors focused on vision; but, I would note the role of the lateral line to the introduction to give a more comprehensive literature review and more integrated into the Discussion.

-Following from that, is there enough contrast in the uCT scans to resolve any of the octavolateral nuclei in the medulla? Given the exploration of the brain in fish selected for polarization, it seems logical to expect differences in the size of the lateralis columns (medial & caudal octavolateral nuclei), which are the sensory nuclei associated with lateral line afferents. And (though it is well outside of the scope of this study) it would be very interesting in future work to assess any differences in the canal/superficial neuromasts between groups.

-Although a personal preference (and perhaps even a requirement of the journal), the last paragraph of the introduction reads as a mix of methods, results, and discussion. I might suggest cutting this down, which would improve the flow of the MS and reduce length.

Methods

-How were fish housed between experiments? Was the number between groups consistent between experiments? Naslund et al. (2019) showed that there are significant correlations between group size and brain morphology; just highlighting this as a potential confounding variable if they were housed in unequal groups

-Although the authors point to a prior publication for the bioimaging, I would appreciate some (brief) detail on the uCT methods used, particularly specimen prep for imaging (ie, was it contrast enhanced) and basic imaging parameters.

-I would like a bit of clarity on what the authors mean by "relative brain region volume" and "relative size", as I got a bit muddled in the use of the term throughout. Is this calculated a proportion of total brain? Is relative size referring to a residual? The authors state (line 592) that they "obtain measurements of whole brain size volume and relative brain region volume in 11 major brain regions," (not absolute size?), later state that "we ran 11 independent LMMs with each region's volume as dependent variable," and then that relative brain region size was incorporated into the Bayesian

model. I think this is just a semantics issue, but it would be useful to clarify when/if absolute brain region volume was used, and how relative brain region size/volume was calculated.

-It would be useful to have a table in the supplementary material that provides morphometrics and raw data for the individual fish. Although I may have missed it, I personally also find a table with the output of the linear models (e.g. fig 3) to be very useful in terms of assessing the rates of ontogenetic growth between brain regions and the rest of the brain in the two selection lines.

-Very Minor point: I queried whether the authors confirmed if there was a correlation between external eye diameter (ie, what is visible) versus true metrics when the eye is dissected from the eye cup. Although in this dataset it almost certainly is, it is an important point of clarity. Further, eye size is very tightly correlated with body size – it may have been too coarse a metric to really identify any differences between selection lines. I may have missed it, but was the allometric relationship between eye size and body size or brain size calculated?

Line 586: To my point above, specimens are listed as “fully-grown” – can the authors provide body morphometrics? Were there any differences in Body mass, TL, or SL between the two lines?

Discussion

Line 334: “However, our results show that directional selection and associated changes in the brain lead to robust behavioral changes across multiple contexts and that it might affect individual ability to efficiently process social information in response to predation. ” I would be very cautious about implying causality; it is actually uncertain which “caused” the other; selecting for a behavior may have inadvertently been selecting for a certain brain area (and vice versa). It’s a bit of a chicken or the egg conundrum, but I would suggest more cautious wording. Rather than “associated changes in the brain lead to robust behavioral changes” I would use “changes in the brain are correlated with behavioral changes.”

Line 378: Awkward wording: “The study of brain anatomy in artificially selected fish allows to study brain function in relation to behaviors that have important implications in the evolution of sociality.” In addition, this study didn’t explore brain function.

Line 393: I would change “eye morphology” to “eye size”, which is really what was assessed

Line 399: “Specifically, representation of the immediate surrounding in the optic tectum is self-centered while the representation is allocentric in the telencephalon.” This may require just a clarification of wording, but egocentric representations (I believe) have also been demonstrated in fishes, associated with subregions of the telencephalon, at least in relation to visual learning in some groups. It might also be worth pointing to the work that explores topographic alignment between the visual system and other sensory systems.

Line 423: The inferior lobe is not part of the medulla, but rather in the diencephalon (specifically hypothalamus). Unless the authors mea

Line 434-435: “...the medulla has a central function the processing of somatosensory signals, with special emphasis in auditory and gustatory signals.” Again, I am surprised to see such a downplay of the mechanosenses and the octaval nuclei involved in processing these cues. I think this needs to be rectified.

Fig 3B, far right segmentation. I appreciate some of the subregions measured are obscured from view; but, the third “semi-segmented” image is very visually confusing. I suggest using the 3D segmentation software to hide the segmentation meshes obscuring the structures of interest, or else make them transparent, so the other nuclei can be resolved through them, Upon first glance, it looked as though the segmentations themselves were done improperly.

Overall, I enjoyed this paper and find it to be a solid contribution to the literature. I am excited by the possibilities this paper provides for future work and feel this is an important way forward for finding links between brain form and behavioral differences.

Reviewer #2:

Remarks to the Author:

This manuscript presents thorough empirical work comprising experiments with individuals and groups of guppies of control and polarisation-selection lines, combined with micro-CT scans of their brains to get further mechanistic insights into the evolution of schooling and coordinated behaviour. It is found that, like in the open field test, groups of polarisation-selected guppies are more polarised than control fish in the context of a novel object and a fake predator, and individuals spending less time inspecting the predator. Neuroanatomical analyses reveal polarisation-selected females have a larger thalamus and optic tectum cups, known to be associated with processing of sensory information, and a smaller medulla oblongata, which the authors suggest may be the result of trade-offs between general and specific sensory capabilities needed for coordinated movement. Finally, tests of guppies' visual capabilities revealed no differences between the selection and control lines, suggesting the observed differences are not associated with differences in visual perception.

I applaud the authors for the impressive amount of work that went into this study, from generating the selection lines to running the large number of schooling experiments, individual experiments on perception, detailed brain scans and neuroanatomical work, and thorough statistical analyses. The manuscript also reads nicely and provides insights into the evolution of schooling behaviour. Together, my take on the findings is that polarisation-selected fish show a universal increase in coordination behaviour (i.e. not context-specific) that this increase in coordination cannot be well explained by improved visual perceptual capabilities, and that it is likely associated with improved processing of (social) information and attention. Although this is largely in line with the authors' interpretations, the manuscript is considerably more focused on the anti-predator aspect and benefits of polarisation. I am not so much in favour of this as it is clear from the results that the fish are generally more polarised and so yes it is interesting that fish stay more polarised even in the context of a (fake) predator (and even when near it or when behind or in front) but that is also the null hypothesis, as the results with the novel object also show. The authors add to this result by investigating inspection behaviour, but here they only randomly focus on one fish in the group, thus disregarding any social mechanisms that may drive any differences, nor have experiments on anti-predator behaviour with individual fish been included, a shortcoming they highlight in the discussion (lines 339-341). I agree with the authors' statement in the discussion on lines 333-335 "our results show that directional selection and associated changes in the brain lead to robust behavioral changes across multiple contexts", but find statements like "our behavioral analyses indicate that rapid evolution of schooling propensity affects how groups of fish behave when encountering a threat" (lines 327-328) are a bit misleading as the results do not provide clear evidence for additional differences related to observed schooling propensity. Furthermore, the experiments on visual perception are interesting but in my opinion not that straightforward to be included in this study by itself, or at least further individual experiments more linked to the functions of the thalamus and optic tectum cups, such as information processing, would be very relevant in my opinion. Taken together, I think this is a nice and valuable study but I am not completely convinced that its insights and broader impact are fit for *Nature Communications*. Below I provide some further general and specific comments that hopefully help you improve your paper.

Like in previous work of many of the authors (Kotrschal et al., 2020), you found here that the polarisation-selected fish were not only more polarised but also moved at a higher speed. Although in

the previous study you showed this was not purely due to selecting faster fish, here your finding of higher polarisation across the contexts and for the subsetted analyses in the context of the fake predator are all likely to do with fish simply moving at higher speeds; low polarisation is not possible at higher speeds. To really understand properly what is going on here, a good first step would be to create heatmaps of group speed versus group polarisation for your selection and control lines in the different contexts (and subsetted datasets). Furthermore, the fact that polarisation-selected fish moved at considerably higher speeds in all contexts, including with the predator model, could potentially also help explain the finding that such fish showed less inspection behaviour: they had less time to inspect as the group tended to be more active. Again, further explorations of your data can give important insights into such potential explanations. A somewhat related but quite different point is that I think it is interesting to better highlight that your polarisation-selected fish were more cohesive despite being faster and more aligned in the open field, behaviours that tend to be negatively correlated (the faster groups move the less cohesive they tend to be). And even though fish become very cohesive in the predator model context, polarisation-selected fish were still considerably polarised.

The group trials were run in water of only 3cm depth. This is very limited and less than any other experimental studies I know on schooling (except your Kotschal et al 2020 paper). The body length and height of the fish is also not provided anywhere, therefore it is not clear how much vertical clearance the fish had. Furthermore, it is questionable how you can run experiments with a coffee cup as novel object and a fake predator in such shallow water as both of them would surely partly stick out above the water.

You use both the terms "polarisation-selected" and "artificial selection for higher schooling propensity" interchangeably across the manuscript but they are not. Please choose one and stick to that. Your selection experiments focused on polarisation so it makes sense to use the former.

Lines 37-39: This statement is too strong. There is quite some empirical work on the fitness and evolution of coordinated movements of grouping animals, including your own work. Perhaps be more specific about what you mean with 'the mechanistic basis' here.

Lines 41-42: rephrase to "presence of a potential threat" or "presence of a fake predator"

Lines 42-44: You find the polarised-selected fish differ in certain brain regions, which is not the same as "showing these behavioural differences are linked to changes in brain regions".

Line 44: Better to clarify you ran further experiments with individual fish to investigate differences in visual capabilities. This is not clear here currently.

Lines 52-64: I would like you to better entertain the idea that coordination and collective motion as such may be largely a result of selection for grouping rather than selected for directly. Information acquisition may be an important factor as well but simple being and moving with others and thereby reducing predation risk and increasing foraging/energetic benefits may be the main driver for emerging collective motion.

Lines 66-68: Same as previous point. To what extent may we see coordinated movement in animal groups simply as a result of animals trying to be near others, and would we then really still expect the brain to play a major role in coordination? I think this would be very species and context specific.

Lines 69-70: Here you are overlooking neurological work studying the onset of schooling in fish

Lines 92-93: See comment above, stick to one term for your selection experiment.

Lines 97-102: Clarify here that the selection lines did not simply select for faster fish, which would also have higher polarisation due to its need to stay together at higher speed

Lines 112-114: Clarify here that those tests were with individual fish

Lines 126-137: Please state the sample size somewhere here in the main text as well as the body size of the fish, which is not provided anywhere as far as I can see.

Lines 145-152: It might be that the groups were already close to the minimum of inter-individual spacing, which Figure 1A suggests, with fish only being 2-3cm from one another. You could consider this possibility in the discussion, which is also interesting given the fact that the polarisation-selected fish manage to stay more polarised than control fish.

Lines 156-158: What about the relationship between speed and polarisation? Please see my main point above with some critical thoughts and suggestions.

Lines 176-182: For these results, and elsewhere, please also give an indication of the effect size in the text, e.g. "control lines spent on average 30% more time inspecting the fake predator than selection lines" [based on Fig 2D]. Also it is not clear here what is "selection: Ratio", is that the score of the selection lines divided by the score of the control lines? This should be clarified.

Lines 172-200: The order of the results presented here is not the same as in the figure; you start with figure 2d-e, then 2c, then 2a and 2b. Therefore either reorder the text or the figure panels so they are in the same order

Lines 210-216: Perhaps it would be good to clarify what areas you focused on and why. Is this more or less all (major) brain areas or did you leave out certain ones? And maybe also consider providing a short description of the key areas' function.

Lines 210-214: Again please include some indication of effect size. From Figure 3 it seems the differences in size of the different regions are not very large?

Figure 3: This figure can be improved by better clarifying the different regions of the brain in figure b, such as with a legend or by writing the abbreviations of the regions on the image, and reflecting the colors in the panels of figure A so that the reader can quickly link a plot to a certain region of the brain. Also, even with the right-most image in B it is still very hard to distinguish the TS, NG, and OTc so I suggest to replace it for a zoomed in image.

Lines 226-235: It is not immediately obvious to me how these additional analyses add to the results given above while they do seem to add a lot of statistical tests. Please better clarify how this result of a negative correlation between the two areas extends beyond finding they were different in the selection lines as well as how maybe with your bayesian approach you could account for type I error by running a large number of correlations.

Line 246: Clarify if these 112 individuals were also used in the schooling experiments and if the experiment was indeed performed after the group trials.

Lines 253-266: This paragraph is missing a straight-to-the point explanation of the test and what it measures. Also a (supplementary) figure of the setup would be helpful.

Lines 274-281: The same. What are you expecting here from the test? You don't explain what outcomes the test actually provides and how. For example, line 281 states the fish followed the direction of the rotating stimulus, but it is not clear to me

Lines 307-309: You cannot make this statement. Your inspection analyses focused on individuals only so you cannot say that fish spent less time inspecting individually VS socially, implied here by stating 'selected groups rely more on neighbour information'. It would have been good to investigate the inspection tendency of all individuals over time, something that is possible with the present data, and thereby actually be able to make a statement as given here.

Lines 309-312: To me it is a bit too far to go from not finding any differences in these abstract tests with moving lines to the interpretation that thus polarisation-selected fish are not better able at distinguishing threats at longer distances or acquire visual information of neighbours. Stick closer to what the tests actually measure and discuss the results as such.

Line 314: Again, here you are talking about selection for higher schooling propensity rather than polarisation

Lines 361-365: This is not clear. What do you mean with 'activity' and how do you mean activity is primarily associated with the exchange of directional information? And why do you interpret finding a difference in activity as suggesting there are differences in the ability to detect neighbour movements or threats? I don't follow these interpretations.

Lines 404-420: As some of the main functions of the thalamus are to relay sensory information, to me it seems more logical to also interpret the enlarged thalamus in polarisation-selected females in the same direction as the enlarged optic tectum cup; that these fish are better able at processing (visual) information (from others) and are thereby able to better align with others while on the move. Currently this section primarily focuses on potential for its role in predator inspection behaviour and aggression, which is less likely in my opinion. Please consider and discuss further.

Lines 431-434: The argument given here is not really correct. That fish in groups startle less than those alone is to do with direct anti-predator benefits of grouping - at the proximate level - while your

observation that polarisation-selected fish show less predator inspection is more at the ultimate level and not much can be said about changes at the proximate level due to trials with individual fish lacking in the present study.

Lines 458-464: These concluding sentences correctly reflect the main findings and interpretation of the study.

Lines 477-479: Also clarify here how the polarization-selected females differed in speed from control lines.

Lines 493-494: You used super shallow water depth of only 3cm, please see my main comment about this.

Lines 499-500: Provide details about the length and height of the model and how it was kept into place.

Lines 497-501: Was the orientation of the mug and predator model always presented in the same orientation or was it randomized? Without it, how can you account for arbitrary spatial preferences in the tank in your analyses, such as those related to front-back position?

Lines 510-511: Please just state the three variables you measured; I suggest to not suddenly start talking about these possible 'axes' of collective motion and polarisation does not necessarily have to be part of a 'sociability axis'

Lines 514-516: In this case your measure of polarisation could be biased as you are losing information on alignment when fish are further away from one another. This is especially critical as polarisation is the main measure of your study and used for creating the selection lines. You need to provide statistics to show the proportion of time groups were beyond the threshold and how this differs between the lines, how your subsetted polarisation measure is related to polarisation without any subsetting, and an explanation why this threshold was used. This is an important point that needs addressing.

Lines 517-518: How often did this happen? Again, it is important you provide the proportion of time you have correct tracking data for and indicate if this was not different between the lines and the different tests. And why did you use a threshold of 16 frames?

Lines 518-520: It is not clear how this is different from that states on 514-516. Do you mean polarisation without subsetting?

Lines 546-553: How much insight can we properly get into inspection behaviour by just looking at the behaviour of one individual in a group of 5? We lose any information about the social aspect of inspection behaviour, which should actually be very interesting in the context of the present work. Is it just that polarised-selected individuals inspect the predator less irrespective of being with others or not or that the evolved differences in social behaviour affect individuals inspection behaviour. From the present analyses this cannot be disentangled.

Lines 592-596: More (background) information should be provided here (or in the main text) about the brain regions. Why did you focus on these 11? Is the reason that together they comprise the majority of the fishes' brain? See also my more general comment.

Lines 626-634: Still a lot of key information is missing here, making it hard to properly understand what was done. So the projection was 360 degrees on all the walls thus surrounding the fish? How did you achieve this? What do you mean with "rotating bands", that the bands move along the walls in circles? And is the stimulus only of a couple bands that thus move around or what do the fish actually see? And foremost, why would the fish respond to one type of width stimulus and not the other, how is that linked to acuity? What do you expect the fish to not see when presenting moving bands?

Line 653: You mean the bands were projected 360 degrees on the walls?

REVIEWER COMMENTS

Below we provide a point-by-point response to reviewers' comments (in bold), including edits in the manuscript driven by these comments (in blue).

Reviewer #1 (Remarks to the Author):

I've read through the MS, "Evolution of schooling propensity in the guppy drives changes in anti-predator behavior that are linked to neuroanatomy," by Corral-Lopez et al, which explored links between neuroanatomy and coordinated movement in guppies. The authors found differences in the superficial layers of the optic tectum, thalamus, and medulla (but not characters related to visual perception) in artificial selection lines of female guppies, selected for divergence in polarization behavior.

Overall, I find this to be an excellent, well written MS that makes important steps forward in the field of comparative neuroanatomy. I find the questions that can be explored with artificial selection lines to be very interesting in terms of closing the gap between neuroanatomy and variation in behavior. My suggestions for improvement are relatively minor, but should be addressed prior to publication

We sincerely thank the reviewer for dedicating their time and expertise to reviewing our manuscript. We greatly appreciate their kind words regarding our work and their valuable suggestions that have contributed to enhancing the quality of the manuscript.

Introduction

-Pg 2, line 57: Note the plural of fish when discussing multiple species is fishes; a few instances of this throughout.

We have carefully revised this issue throughout the manuscript.

-I was surprised to see an almost exclusive focus on vision in relation to coordinated movement, with very little mention of mechanosensory cues (until the very last paragraph of the Discussion). While vision has definitely been shown to play a role in coordinated motion, it seems a bit of a glaring gap to neglect mention of hydrodynamic cues in schooling behavior until the end. I don't think it's a strong critique that the authors focused on vision; but, I would note the role of the lateral line to the introduction to give a more comprehensive literature review and more integrated into the Discussion.

We have now noted the role of the lateral line in the introduction and integrated it more in the discussion:

Intro (Lines 76-81):

'Collective motion has evolved multiple times in fishes and is widely understood as a behavioral adaptation aimed at reducing the risk of predation². This behavioral adaptation is underpinned by the efficient acquisition of information from external cues through visual and lateral line sensory systems³⁻⁶.'

Discussion (lines 496-508):

'The medulla oblongata is an important relay center of nervous signals between the spinal cord and ascendant brain regions with well-described functions in teleosts. This region has an important role in basic motor function through processing of mechanosensory stimuli from hydrodynamic information⁶¹. Consistently found across fish species, the lateral line nerves terminate in the medial and caudal octavo lateralis nucleus in the medulla⁶². As the lateral line is crucial for schooling through cues that allow fish to assess neighbor changes in speed and direction, the reduced relative size of the medulla in polarization-selected lines could be

associated to potential differences in the ability to integrate and process information received through these nerves. Yet, our previous studies evaluating motor control capabilities found no difference between polarization-selected and control female guppies^{21,63}. Further studies evaluating information processing of mechanosensory input in these selection lines is nonetheless paramount, with special focus on low light and high turbidity conditions.'

-Following from that, is there enough contrast in the uCT scans to resolve any of the octavolateral nuclei in the medulla? Given the exploration of the brain in fish selected for polarization, it seems logical to expect differences in the size of the lateralis columns (medial & caudal octavolateral nuclei), which are the sensory nuclei associated with lateral line afferents. And (though it is well outside of the scope of this study) it would be very interesting in future work to assess any differences in the canal/superficial neuromasts between groups.

Thank you for this suggestion and those nuclei would be exciting to explore further. Unfortunately, with our current resolution this is not possible. We will however gladly keep this suggestion in mind for future work incorporating classic histological approaches.

-Although a personal preference (and perhaps even a requirement of the journal), the last paragraph of the introduction reads as a mix of methods, results, and discussion. I might suggest cutting this down, which would improve the flow of the MS and reduce length.

We have now removed reference to results that are presented in detail in the next section following this final paragraph in the introduction. It now reads like this (lines 124-134):

'We use these lines to investigate potential changes in the visual system and anti-predator behavior following directional selection for polarization, as well as to study the association between increased schooling propensity and changes in brain anatomy. For this, we performed a series of experiments and measurements to: i) evaluate collective motion patterns and predator inspection of individuals from polarization-selected and control female guppies when swimming in groups, ii) quantify brain region sizes with microcomputed tomography (microCT) of female guppies from polarization-selected and control lines, and iii) perform comprehensive tests of visual capabilities of these fish, spanning eye size, visual acuity and temporal resolution measurements of individual fish. Through the study of collective behaviour in an ecologically relevant setting and how selection for higher polarization is associated to brain structure size variation we identify potential evolutionary pathways leading to collective motion.'

Methods

-How were fish housed between experiments? Was the number between groups consistent between experiments? Naslund et al. (2019) showed that there are significant correlations between group size and brain morphology; just highlighting this as a potential confounding variable if they were housed in unequal groups

We have clarified in the text that all fish were kept in single-sex groups of eight individuals prior to experiments and morphological measurements (Lines 560-564):

'All fish were removed from their parental tanks after birth, separated by sex at the first onset of sexual maturation, and afterwards transferred to single-sex groups of eight individuals in seven liter tanks. We kept all fish used for anti-predator response experiments, visual capacity tests and brain morphology measurements in these groups throughout their life span. Fish were not reused for experiments across these three categories'

-Although the authors point to a prior publication for the bioimaging, I would appreciate some (brief) detail on the

uCT methods used, particularly specimen prep for imaging (ie, was it contrast enhanced) and basic imaging parameters.

We have now added information on sample preparation and imaging parameters for microCT scanning in the methods (lines 682-691):

‘We euthanized animals with an overdose of benzocaine and fixated the whole fish in 2% glutaraldehyde and 4% paraformaldehyde in phosphate buffered saline (PBS) for five days. Following two PBS washes, the brains were dissected out and stained for 48 h in 1% osmiumtetroxide. We embedded the stained brains in 3% agar and scanned them using microcomputed tomography (microCT, Skyscan 1172, Bruker microCT, Kontich, Belgium). The scanner operated at a voltage of 80 kV, a current of 125 μ m, with a 0.5 mm aluminum filter. Images were acquired using an isotropic pixel size of 2.4 μ m. We reconstructed cross-sections from scanned images following a in NRecon (Bruker microCT) following a protocol successfully implemented in a previous study evaluating neuroanatomical differences between guppies up- and down-selected for relative brain size⁵³.’

-I would like a bit of clarity on what the authors mean by “relative brain region volume” and “relative size”, as I got a bit muddled in the use of the term throughout. Is this calculated a proportion of total brain? Is relative size referring to a residual? The authors state (line 592) that they “obtain measurements of whole brain size volume and relative brain region volume in 11 major brain regions,” (not absolute size?), later state that “we ran 11 independent LMMs with each region’s volume as dependent variable,” and then that relative brain region size was incorporated into the Bayesian model. I think this is just a semantics issue, but it would be useful to clarify when/if absolute brain region volume was used, and how relative brain region size/volume was calculated.

To measure differences in relative brain size between polarization-selected and control fish we included body size as a covariate in our statistical models. To measure differences in relative brain region size between polarization-selected and control fish we included whole brain size volume (minus the volume of the region of interest) as a covariate in statistical models. We have now clarified this in our statistical analyses’ description (Lines 701-713):

‘We tested for overall differences in relative brain size between polarization-selected and control lines using a linear mixed model (LMM) with brain volume as dependent variable, body size (standard length) as covariate, selection regime as fixed effect, and replicate as random effect. For each brain region, we used two different approaches to determine whether polarization-selected and control lines differ in relative brain region size. First, we ran 11 independent LMMs with each region’s volume as dependent variable, whole brain volume (excluding the volume of the region of interest) as covariate, selection regime as fixed effect, and replicate as random effect. LMMs were run in R (v 4.1.3) using lme4 and lmerTest packages^{71,72}. Second, to take into consideration that brain region volumes may be interdependent, we used a more conservative approach and analyzed the data using a Bayesian multilevel model that included 11 brain regions as dependent variables in a fully multivariate context. The full model included an analogous structure to those used in the independent LMMs for each brain region.’

We have additionally clarified this in the results description (lines 255-264):

‘Polarization-selected and control fish showed no differences in relative brain size (whole brain volume in relation to their body size; $LMM_{\text{relativebrain}}$: $t = -0.41$, $df = 23.29$, $p = 0.682$; Fig. 4a). However, analyses of relative brain region size (volume of each region in relation to the volume of the rest of the brain) indicated that the thalamus and optic tectum cups are approximately 7% and 4% larger in polarization-selected than control females respectively (LMM_{thalamus} : Odds Ratio (OR)_{polarization} = 0.929 (0.866-0.998); $t = 2.187$, $df = 25$, $p = 0.038$; $LMM_{\text{o.tectum}}$: Odds Ratio (OR)_{polarization} = 0.959 (0.924-0.994); $t = 2.409$, $df = 23.09$, $p = 0.024$; Fig. 4a), and the medulla oblongata is an 8% larger in control females (LMM_{medulla} : Odds Ratio (OR)_{polarization} = 1.08 (1.01-1.14); $t = -2.65$, $df = 23.91$, $p = 0.013$; Fig. 4a).’

-It would be useful to have a table in the supplementary material that provides morphometrics and raw data for the individual fish. Although I may have missed it, I personally also find a table with the output of the linear models (e.g. fig 3) to be very useful in terms of assessing the rates of ontogenetic growth between brain regions and the rest of the brain in the two selection lines.

Together with the manuscript we will be providing all data files and scripts used to calculate the results presented. The data file with brain volumes, body sizes and brain region volumes for all fish will be provided and will hence be available to assess this information. Apologies if we have misunderstood the reviewer’s suggestion, but we are unsure whether the additional presentation of that data in a Supplementary Table is a better format than providing access to the raw data. Of course, we are more than happy to reconsider if we have misunderstood the comment/or reviewers and editor agree it would be beneficial to add also an additional table in the Suppl. Mat.

As for the output of linear models, the information is provided in supplementary tables 6 and 7:

‘Table S6. Results from independent Linear Mixed Models evaluating differences in relative brain and relative brain region size between polarization-selected and control female guppies.

Region	Coefficient	Estimate	SE	df	t	P-value
Whole brain	Intercept	10.706	0.929	22.992	11.520	< 0.001
	Sel. Line (Polarization)	-0.015	0.037	23.296	-0.415	0.682
	Log (SL)	1.581	0.292	22.899	5.401	< 0.001
Olfactory bulbs	Intercept	-5.639	4.048	24.311	-1.393	0.176
	Sel. Line (Polarization)	-0.054	0.071	23.443	-0.759	0.455
	Log (rest of brain)	0.973	0.257	24.319	3.782	< 0.001
Ventral telencephalon	Intercept	-7.112	4.022	23.970	-1.768	0.0897
	Sel. Line (Polarization)	-0.054	0.070	23.366	-0.762	0.453
	Log (rest of brain)	1.107	0.255	23.968	4.329	< 0.001
Dorsal telencephalon	Intercept	-3.002	1.502	25.000	-1.998	0.0567
	Sel. Line (Polarization)	0.029	0.027	25.000	1.096	0.2837
	Log (rest of brain)	0.935	0.095	25.000	9.780	< 0.001
Thalamus	Intercept	-1.019	1.805	25.000	-0.565	0.577
	Sel. Line (Polarization)	0.073	0.033	25.000	2.187	0.038
	Log (rest of brain)	0.950	0.115	25.000	8.197	< 0.001
Hypothalamus	Intercept	-3.858	2.745	25.000	-1.405	0.172
	Sel. Line (Polarization)	0.048	0.049	25.000	0.974	0.339
	Log (rest of brain)	1.098	0.175	25.000	6.259	< 0.001
Nucleus glomerulus	Intercept	-2.352	9.716	25.000	-0.242	0.811
	Sel. Line (Polarization)	-0.067	0.176	25.000	-0.381	0.706
	Log (rest of brain)	0.807	0.617	25.000	1.308	0.203
Torus semicircularis	Intercept	-0.665	2.368	24.977	-0.281	0.781
	Sel. Line (Polarization)	-0.0325	0.042	24.106	-0.758	0.456
	Log (rest of brain)	0.829	0.150	24.982	5.496	< 0.001
Optic tectum cup	Intercept	1.073	0.966	23.494	1.111	0.2780
	Sel. Line (Polarization)	0.042	0.017	23.095	2.409	0.024
	Log (rest of brain)	0.800	0.061	23.474	12.933	< 0.001
Optic tectum core	Intercept	-2.883	1.658	25.000	-1.739	0.0944
	Sel. Line (Polarization)	0.006	0.030	25.000	0.211	0.8347
	Log (rest of brain)	1.040	0.106	25.000	9.805	< 0.001
Cerebellum	Intercept	-0.403	1.458	24.350	-0.277	0.784
	Sel. Line (Polarization)	-0.007	0.026	23.221	-0.275	0.786
	Log (rest of brain)	0.908	0.093	24.358	9.703	< 0.001
Medulla oblongata	Intercept	-0.145	1.519	24.920	-0.096	0.9246
	Sel. Line (Polarization)	-0.074	0.028	23.916	-2.656	0.013
	Log (rest of brain)	0.969	0.099	24.935	9.761	< 0.001

Table S7a. Results from a Bayesian multilevel model evaluating differences in relative brain region size between polarization-selected and control female guppies. Stars indicate estimates that do not include zero in the confidence interval range based on the posterior samples drawn from the model.’

Covariate	Estimate	Est.Error	l.95..CI	u.95..CI	
Medulla_Intercept	0.19	0.32	-0.37	0.77	
Cerebellum_Intercept	0.05	0.50	-1.08	1.04	
Nucleus glomerulus_Intercept	0.09	0.48	-0.84	1.04	
Torus semicircularis_Intercept	0.14	0.44	-0.69	1.03	
Thalamus_Intercept	-0.25	0.34	-0.97	0.31	
Optic tectum cups_Intercept	-0.19	0.51	-1.26	0.89	
Hypothalamus_Intercept	-0.06	0.39	-0.84	0.73	
Olfactory bulbs_Intercept	0.13	0.54	-0.94	1.26	
Ventral telencephalon_Intercept	0.05	0.63	-1.23	1.43	
Dorsal telencephalon_Intercept	-0.09	0.32	-0.72	0.56	
Optic tectum core_Intercept	-0.02	0.28	-0.56	0.54	
Medulla oblongata_Selection	-0.42	0.18	-0.79	-0.06	*
Medulla oblongata_Rest of the brain	0.91	0.09	0.74	1.09	*
Cerebellum_Selection	-0.07	0.21	-0.48	0.35	
Cerebellum_Rest of the brain	0.96	0.11	0.75	1.18	*
Nucleus glomerulus_Selection	-0.21	0.40	-0.98	0.58	
Nucleus glomerulus_Rest of the brain	0.34	0.20	-0.06	0.75	
Torus semicircularis_Selection	-0.24	0.32	-0.87	0.38	
Torus semicircularis_Rest of the brain	0.72	0.16	0.40	1.04	*
Thalamus_Selection	0.49	0.23	0.04	0.94	*
Thalamus_Rest of the brain	0.92	0.11	0.70	1.16	*
Optic tectum cups_Selection	0.32	0.16	0.00	0.63	*
Optic tectum cups_Rest of the brain	0.88	0.08	0.72	1.06	*
Hypothalamus_Selection	0.12	0.26	-0.39	0.64	
Hypothalamus_Rest of the brain	0.87	0.13	0.61	1.14	*
Olfactory bulbs_Selection	-0.27	0.33	-0.93	0.38	
Olfactory bulbs_Rest of the brain	0.62	0.17	0.29	0.95	*

Ventral telencephalon_Selection	-0.17	0.31	-0.76	0.43	
Ventral telencephalon_Rest of the brain	0.58	0.16	0.27	0.89	*
Dorsal telencephalon_Selection	0.18	0.21	-0.25	0.60	
Dorsal telencephalon_Rest of the brain	0.91	0.11	0.69	1.13	*
Optic tectum core_Selection	0.05	0.20	-0.34	0.44	
Optic tectum core_Rest of the brain	0.93	0.10	0.73	1.13	*

-Very Minor point: I queried whether the authors confirmed if there was a correlation between external eye diameter (ie, what is visible) versus true metrics when the eye is dissected from the eye cup. Although in this dataset it almost certainly is, it is an important point of clarity. Further, eye size is very tightly correlated with body size – it may have been too coarse a metric to really identify any differences between selection lines. I may have missed it, but was the allometric relationship between eye size and body size or brain size calculated?

We unfortunately cannot provide further information between eye diameter and true metrics from eyes dissected or between brain morphology and eye size in the subset of fish used in this study. However, our previous work directly evaluated this relationship using guppies artificially selected for large and small relative brain size (Corral-Lopez et al. 2017) and identified no differences in body size between polarization-selected and control lines fish (Kotrschal, Szorkovsky et al. 2020). Additionally, we found no differences in body size between the subset of polarization-selected and control fish randomly chosen to perform vision performance tests and eye size measurements. We now include new analyses confirming the well-established positive correlation between body size and eye size in this species and confirming that artificial selection for social coordination did not alter this correlation (Lines 300-308):

‘In our study, we found no difference between polarization-selected and control lines in either absolute eye size or relative eye size, the proportional size of the eye in relation to body size (Fig. 5A; LMM_{eye size}: Estimate_{selection} = -0.03 [-0.24, 0.19], t = -0.52, df = 2, p = 0.66; LMM_{relative eye size}: Estimate_{selection} = -0.00 [-0.01, 0.01], t = -0.13, df = 2, p = 0.90; Supplementary Table 8). As previously reported in guppies²⁸, we found a strong positive correlation between eye size and body size (Pearson’s correlation test: t = 11.72, df = 110, p < 0.001, cor [95% CIs] = 0.74 [0.65, 0.82]), but this correlation did not differ between polarization-selected and control lines (LMM_{eye size}: Estimate_{selection*body size} = 0.010 [-0.016, 0.039], t = 0.75, df = 101, p = 0.45).’

References cited:

Corral-López, A., Garate-Olaizola, M., Buechel, S. D., Kolm, N. & Kotrschal, A. On the role of body size, brain size, and eye size in visual acuity. *Behav Ecol Sociobiol* **71**, 1–10 (2017).

Kotrschal, A. et al. Rapid evolution of coordinated and collective movement in response to artificial selection. *Sci Adv* **6**, eaba3148 (2020).

Line 586: To my point above, specimens are listed as “fully-grown” – can the authors provide body morphometrics? Were there any differences in Body mass, TL, or SL between the two lines?

There were no statistically significant differences in body size between polarization-selected and control fish used for brain morphology measurements, visual performance

tests and collective motion tests. We have now included these body size measurements in the manuscript (lines 573-576; 682-685; 732-734):

‘We evaluated anti-predator behavior in polarization-selected and control female guppies by conducting assays on 164 groups of eight fish in white arenas with 55 cm diameter and 3 cm water depth (polarization-selected: n = 83, standard length mean [95% CIs] = 248 mm [240, 255]); control: n = 81, standard length mean [95% CIs] = 245 mm [238, 253]).’

‘We assessed neuroanatomical features of polarization-selected and control F3 females of approximately 6 months old (polarization-selected: n = 15, standard length mean [95% CIs] = 238 mm [235, 242]; control: n = 15, standard length mean [95% CIs] = 237 mm [232, 241]).’

‘We evaluated the ability to perceive detail (visual acuity) in 9-12 months old female guppies (polarization-selected: n = 59, standard length [95% CIs] = 270 mm [266, 274], control: n = 57, standard length [95% CIs] = 271 mm [268, 275]).’

Discussion

Line 334: “However, our results show that directional selection and associated changes in the brain lead to robust behavioral changes across multiple contexts and that it might affect individual ability to efficiently process social information in response to predation. “ I would be very cautious about implying causality; it is actually uncertain which “caused” the other; selecting for a behavior may have inadvertently been selecting for a certain brain area (and vice versa). It’s a bit of a chicken or the egg conundrum, but I would suggest more cautious wording. Rather than “associated changes in the brain lead to robust behavioral changes” I would use “changes in the brain are correlated with behavioral changes.”

We have revised accordingly (Lines 411-414):

‘However, our results show that directional selection and associated changes in the brain are correlated to behavioral changes in the presence of a threat and that it might affect individual ability to acquire private information about a potential predator.’

Line 378: Awkward wording: “The study of brain anatomy in artificially selected fish allows to study brain function in relation to behaviors that have important implications in the evolution of sociality.” In addition, this study didn’t explore brain function.

We have now removed this sentence from the manuscript.

Line 393: I would change “eye morphology” to “eye size”, which is really what was assessed

We have changed accordingly in all instances where “eye morphology” was used.

Line 399: “Specifically, representation of the immediate surrounding in the optic tectum is self-centered while the representation is allocentric in the telencephalon.” This may require just a clarification of wording, but egocentric representations (I believe) have also been demonstrated in fishes, associated with subregions of the telencephalon, at least in relation to visual learning in some groups. It might also be worth pointing to the work that explores topographic alignment between the visual system and other sensory systems.

We have revised accordingly and removed references to self-centered representation in the optic tectum (Lines 460-466):

‘Together, these findings suggest that differences found between polarization-selected and control lines in this section of the optic tectum should have an effect in their ability to control body orientation during complex social maneuvers like predator evasion or swimming with low inter-individual distances, but not in sensory information acquisition. This is consistent with current considerations of the tectum as more than a visuomotor relay, with implementation

of intrinsic processing of visual information to extract critical features and send processed outputs to the posterior thalamus and then to further decision-making regions of the brain⁵¹.’

It might also be worth pointing to the work that explores topographic alignment between the visual system and other sensory systems.

Following changes in our discussion on brain morphology and collective behavior, we now focus on the potential role of a larger tectum and thalamus as facilitators of an efficient relay of processed information into decision-making regions of the brain, as well as the potential effect of a reduced medulla in potential trade-offs between vision and other sensory and mechanosensory systems.

Line 423: The inferior lobe is not part of the medulla, but rather in the diencephalon (specifically hypothalamus). Unless the authors mea

While the comment from the reviewer was unfortunately incomplete, changes in the discussion in this new version included the removal of references to the inferior lobe.

Line 434-435: "...the medulla has a central function the processing of somatosensory signals, with special emphasis in auditory and gustatory signals." Again, I am surprised to see such a downplay of the mechanosenses and the octaval nuclei involved in processing these cues. I think this needs to be rectified.

Following the reviewer suggestions, our discussion includes now references of the mechanosenses and the octaval nuclei involved in processing these cues (Lines 495-508):

‘In contrast to the thalamus and the optic tectum cup, we found that the medulla oblongata was smaller in polarization-selected lines. The medulla oblongata is an important relay center of nervous signals between the spinal cord and ascendant brain regions with well-described functions in teleosts. First, it has an important role in basic motor function through processing of mechanosensory stimuli from hydrodynamic information⁶¹. Consistently found across fish species, the lateral line nerves terminate in the medial and caudal octavo lateralis nucleus in the medulla⁶². As the lateral line is crucial for schooling through cues that allow fishes to assess neighbor changes in speed and direction, the reduced relative size of the medulla in polarization-selected lines could be associated to potential differences in the ability to integrate and process information received through these nerves. Yet, our previous studies evaluating motor control capabilities found no difference between polarization-selected and control female guppies^{21,63}. Further studies evaluating information processing of mechanosensory input in these selection lines is nonetheless paramount, with special focus on low light and high turbidity conditions.’

Fig 3B, far right segmentation. I appreciate some of the subregions measured are obscured from view; but, the third "semi-segmented" image is very visually confusing. I suggest using the 3D segmentation software to hide the segmentation meshes obscuring the structures of interest, or else make them transparent, so the other nuclei can be resolved through them, Upon first glance, it looked as though the segmentations themselves were done improperly.

These pictures are from an earlier study in which we developed the guppy brain atlas (Kotrschal et al. 2017). While we can still work with the brain atlas, we have no more access to the visualization software and unfortunately cannot edit this figure further. This partially segmented picture was captured from a video we took during the segmentation

process (that is also why some regions are not perfectly defined). As we agree that the partially segmented figure may seem confusing (and does not completely reflect our segmentation) we have removed it from the panel. We have additionally added text legends with abbreviations in each visible brain region with nomenclature used in the y axis of Panel A to facilitate readers to link each plot to the region of interest:

‘Fig 4. The effect of artificial selection for higher polarization in neuroanatomical allometric relationships. (A) The top left panel shows the allometric relationship between whole brain size volume and standard length of the fish (SL). Remaining panels show the relationship between each separate brain region with the rest of the brain ordered rostrally to caudally. Asterisks indicate brain regions with non-overlapping confidence intervals between polarization-selected females (pink; $n = 13$) and control females (blue; $n = 15$) in two consistent statistical analyses (Supplementary Tables 6-7). **(B)** Reconstructed brain regions from microCT - scanned guppy brains. A dorsal (left) and lateral (right) view of a guppy brain with the major brain regions color coded: olfactory bulbs (OB; dark blue), dorsal telencephalon (DT; red), ventral telencephalon (VT; light green), optic tectum (OT; yellow), hypothalamus (Hyp; turquoise), thalamus (Th; purple), cerebellum (Cb; brown), medulla oblongata (MO; dark green). Not visible but measured brain regions include: torus semicircularis, nucleus glomerulus and optic tectum core.’

Reference cited:

Kotrschal, Alexander, et al. "Evolution of brain region volumes during artificial selection for relative brain size." *Evolution* 71.12 (2017): 2942-2951.

Overall, I enjoyed this paper and find it to be a solid contribution to the literature. I am excited by the possibilities

this paper provides for future work and feel this is an important way forward for finding links between brain form and behavioral differences.

We are grateful to the reviewer for their extremely important contributions in improving the manuscript and we also look forward to potential future work combining collective motion and neuroanatomy that can be derived from our study.

Reviewer #2 (Remarks to the Author):

This manuscript presents thorough empirical work comprising experiments with individuals and groups of guppies of control and polarisation-selection lines, combined with micro-CT scans of their brains to get further mechanistic insights into the evolution of schooling and coordinated behaviour. It is found that, like in the open field test, groups of polarisation-selected guppies are more polarised than control fish in the context of a novel object and a fake predator, and individuals spending less time inspecting the predator. Neuroanatomical analyses reveal polarisation-selected females have a larger thalamus and optic tectum cups, known to be associated with processing of sensory information, and a smaller medulla oblongata, which the authors suggest may be the result of trade-offs between general and specific sensory capabilities needed for coordinated movement. Finally, tests of guppies' visual capabilities revealed no differences between the selection and control lines, suggesting the observed differences are not associated with differences in visual perception.

I applaud the authors for the impressive amount of work that went into this study, from generating the selection lines to running the large number of schooling experiments, individual experiments on perception, detailed brain scans and neuroanatomical work, and thorough statistical analyses. The manuscript also reads nicely and provides insights into the evolution of schooling behaviour. Together, my take on the findings is that polarisation-selected fish show a universal increase in coordination behaviour (i.e. not context-specific) that this increase in coordination cannot be well explained by improved visual perceptual capabilities, and that it is likely associated with improved processing of (social) information and attention. Although this is largely in line with the authors' interpretations, the manuscript is considerably more focused on the anti-predator aspect and benefits of polarisation. I am not so much in favour of this as it is clear from the results that the fish are generally more polarised and so yes it is interesting that fish stay more polarised even in the context of a (fake) predator (and even when near it or when behind or in front) but that is also the null hypothesis, as the results with the novel object also show. The authors add to this result by investigating inspection behaviour, but here they only randomly focus on one fish in the group, thus disregarding any social mechanisms that may drive any differences, nor have experiments on anti-predator behaviour with individual fish been included, a shortcoming they highlight in the discussion (lines 339-341). I agree with the authors' statement in the discussion on lines 333-335 "our results show that directional selection and associated changes in the brain lead to robust behavioral changes across multiple contexts", but find statements like "our behavioral analyses indicate that rapid evolution of schooling propensity affects how groups of fish behave when encountering a threat" (lines 327-328) are a bit misleading as the results do not provide clear evidence for additional differences related to observed schooling propensity. Furthermore, the experiments on visual perception are interesting but in my opinion not that straight forward to be included in this study by itself, or at least further individual experiments more linked to the functions of the thalamus and optic tectum cups, such as information processing, would be very relevant in my opinion. Taken together, I think this is a nice and valuable study but I am not completely convinced that its insights and broader impact are fit for Nature Communications. Below I provide some further general and specific comments that hopefully help you improve your paper.

We thank the reviewer for acknowledging the amount of work and the combination of techniques and experiments to generate the study. And also for the kind words on the writing and the insights that the study provides. We are likewise very grateful to the reviewer for the detailed revision and insightful comments. We are overall in agreement with the reviewer's interpretation of the results and we have shifted the focus away from differences observed in anti-predator contexts as exemplified by the revised title:

'Changes driven by the evolution of higher group polarization in the guppy are consistent across different predation pressures and associated with neuroanatomical changes'

We agree that our results suggest further investigations on information processing and direct evaluations of the roles of thalamus and optic tectum cups in social information processing are an important future direction. Yet, we believe it is important to keep results

from experiments on visual performance in combination with anti-predator behavior and polarization across multiple contexts. Also, we realize we may not have been as clear as we might have wished about the methods we used and their importance to the biological relevancy of our visual perception measurements. We have endeavored to revise these sections in order to remove any lingering doubts the reviewer might have about the importance of our results. We are confident that these results provide a good proxy to assess whether the observed differences in polarization might be caused by differences in visual acquirement of information on neighbour movements or their ability to capture sensory information on potential threats at longer distances.

Like in previous work of many of the authors (Kotrschal et al., 2020), you found here that the polarisation-selected fish were not only more polarised but also moved at a higher speed. Although in the previous study you showed this was not purely due to selecting faster fish, here your finding of higher polarisation across the contexts and for the subsetted analyses in the context of the fake predator are all likely to do with fish simply moving at higher speeds; low polarisation is not possible at higher speeds. To really understand properly what is going on here, a good first step would be to create heatmaps of group speed versus group polarisation for your selection and control lines in the different contexts (and subsetted datasets). Furthermore, the fact that polarisation-selected fish moved at considerably higher speeds in all contexts, including with the predator model, could potentially also help explain the finding that such fish showed less inspection behaviour: they had less time to inspect as the group tended to be more active. Again, further explorations of your data can give important insights into such potential explanations. A somewhat related but quite different point is that I think it is interesting to better highlight that your polarisation-selected fish were more cohesive despite being faster and more aligned in the open field, behaviours that tend to be negatively correlated (the faster groups move the less cohesive they tend to be). And even though fish become very cohesive in the predator model context, polarisation-selected fish were still considerably polarised.

We have now created heatmaps of group speed versus group polarizations across contexts, added figure 2 to the manuscript with these findings and incorporated explanations of the role of speed in results and discussion:

Results (Lines 192-210):

‘Correlation between group polarization and speed

To further characterize potential differences in how information spreads in polarization-selected and control fish groups we assessed the effect of group speed in the alignment of fish when exposed to a predation threat. For this, we generated heatmaps showing the correlation of group speed and group polarization across treatments for our different selection lines (Fig. 2). Heatmaps indicated that group polarization of polarization-selected fish in OFTs was constantly close to maximal values of one and differences with control groups were partially independent of the speed of the group (despite larger values of median speed observed in relation to those in control groups). Overall group polarization reduction in the presence of a novel object and a predator model in relation to OFTs were likely associated with a strong reduction in the speed of the group, including periods of no movement of the group. Periods of no movement in response to a threat were observed more frequently in control groups suggesting that they might be an important driver of group polarization differences observed between polarization-selected and control groups. In addition to differences driven by lack of movement, correlations of group polarization and speed indicated that differences between polarization-selected and control groups observed during collective motion resembling motion in OFT’s (mean group speed range 2 – 5 mm/frame) were maintained in the presence of stimuli in the experimental arena (Fig. 2).’

Fig 2. Correlation between group polarization and speed. Heatmaps showing the association between group polarization and speed in control (top) and polarization-selected (bottom) groups of eight guppy females exposed to open field, novel object and predator model assays. Group speed depicts the mean median swimming speed of the individuals in a group in mm per frame. Heatmaps indicate that the overall decrease in polarization in the presence of a novel object and a predator model was associated with a reduction in speed (higher relative density of group speed < 2 mm/frame). The consistent differences in polarization across treatments between polarization-selected and control groups seem to be driven by: i) lower reduction in mean group speed in polarization-selected groups, including periods of no movement (area delimited by blue arrows); and ii) consistent differences in motion characteristics regardless of the presence of stimuli in the assay (lower range of group polarization at group speed >2 mm/frame, area delimited by red arrows).

Discussion (Lines 391-404; 430-434):

‘Speed plays a prominent role in structuring fish schools, as individuals adjust their distance and alignment with neighbours through changes in speed^{21,37,38}. Indeed, empirical studies across several species, as well as modelling approaches, have recently demonstrated a positive correlation between individual’s speed and group polarization³⁸⁻⁴³. In our experiments, the presence of a novel object and a predator model prompted strong reductions in the median speed of individuals in the test. Yet, similar differences between polarization-selected and control females in individual’s median speed and polarization were maintained across all contexts. Swimming performance tests previously performed in these fish demonstrated that selection did not target fish with better physiological capacities to swim faster, but those showing higher speed in response to neighbor positions²¹. Taken together, our results from these selection lines suggest that selection for coordinated movement is largely mediated by

individuals' speed, and that group coordination characteristics such as polarization can emerge from individuals' tendencies to group with conspecifics and fitness benefits acquired from such tendencies at specific contexts.'

'In sticklebacks less sociable individuals are more likely to discover food first in a foraging context³⁷. This agrees with our results in experiments with a predator model, where faster individuals' speed and higher tendencies to group with conspecifics seem to drive the observed lower time inspecting the predator model in the arena and acquire private information on a potential threat.'

The group trials were run in water of only 3cm depth. This is very limited and less than any other experimental studies I know on schooling (except your Kotschal et al 2020 paper). The body length and height of the fish is also not provided anywhere, therefore it is not clear how much vertical clearance the fish had. Furthermore, it is questionable how you can run experiments with a coffee cup as novel object and a fake predator in such shallow water as both of them would surely partly stick out above the water.

With respect, we don't think the shallow water depth in our experimental setup, used to minimize tracking errors from video recordings, is very different from standards in the field taking guppy biology into consideration. From our experience performing behavioral and cognition experiments in guppies and schooling quantification across several species of fishes there are no indications of abnormal behavior in our experimental setup. In our experience performing field collections of guppies in natural habitats, it is common to find streams with similar water depth in which guppies inhabit. We additionally performed a non-exhaustive search for work from other authors and found examples of similar water depths in guppy experimental setups: Evans and Magurran (2000): schooling behavior - 2.5 cm; Ghalambor et al. (2004): swimming performance and escape response - 3 cm; Feyten et al. (2021): decision-making in relation to predation - 3 cm.

The lure fish used as a predator model did not stick out of the water. We have now added information of average female guppy body size and predator model to clarify this. The coffee cup did, but we are confident that it is a valid control between a non-threatening novel object and a potential predator threat. This is so because we have previously found strong differences in the behavioral responses to the predator model in a previous experiment with exact same setup and water depth (Corral-Lopez et al. 2021), and differences observed in this experiment resemble those using a higher water depth in guppies artificially selected for relative brain size (van der Bijl et al. 2015).

References in this response:

van der Bijl, W., Thyselius, M., Kotschal, A., & Kolm, N. (2015). Brain size affects the behavioural response to predators in female guppies (*Poecilia reticulata*). *Proceedings of the Royal Society B: Biological Sciences*, 282(1812), 20151132.

Corral-López, A., Romensky, M., Kotschal, A., Buechel, S. D., & Kolm, N. (2021). Brain size affects responsiveness in mating behaviour to variation in predation pressure and sex ratio. *Journal of evolutionary biology*, 33(2), 165-177.

Evans, J. P., & Magurran, A. E. (2000). Multiple benefits of multiple mating in guppies. *Proceedings of the National Academy of Sciences*, 97(18), 10074-10076.

Ghalambor, C. K., Reznick, D. N., & Walker, J. A. (2004). Constraints on adaptive evolution: the functional trade-off between reproduction and fast-start swimming performance in the Trinidadian guppy (*Poecilia reticulata*). *The American Naturalist*, 164(1), 38-50.

Feyten, L. E., Crane, A. L., Ramnarine, I. W., & Brown, G. E. (2021). Predation risk shapes the use of conflicting personal risk and social safety information in guppies. *Behavioral Ecology*, 32(6), 1296-1305.

You use both the terms "polarisation-selected" and "artificial selection for higher schooling propensity" interchangeably across the manuscript but they are not. Please choose one and stick to that. Your selection experiments focused on polarisation so it makes sense to use the former.

We have changed the text accordingly and now use “polarization-selected” consistently in the manuscript.

Lines 37-39: This statement is too strong. There is quite some empirical work on the fitness and evolution of coordinated movements of grouping animals, including your own work. Perhaps be more specific about what you mean with 'the mechanistic basis' here.

We have clarified this statement (Lines 40-42):

‘However, elucidating the neural mechanisms underlying the evolution of collective behaviors, as well as their fitness effects, remains challenging.’

Lines 41-42: rephrase to "presence of a potential threat" or "presence of a fake predator"

Changed accordingly.

Lines 42-44: You find the polarised-selected fish differ in certain brain regions, which is not the same as "showing these behavioural differences are linked to changes in brain regions".

We have clarified this statement (Lines 44-47):

‘Neuroanatomical measurements of polarization-selected individuals indicated changes in brain regions previously suggested to be important regulators of perception, fear and attention, and motor response.’

Line 44: Better to clarify you ran further experiments with individual fish to investigate differences in visual capabilities. This is not clear here currently.

We have changed accordingly (Lines 46-51):

‘Additional visual acuity and temporal resolution tests performed in polarization-selected and control individuals indicate that observed differences in anti-predator and schooling behavior should not be attributable to changes in visual perception, but likely to more efficient relay of sensory input in the brain of polarization-selected fish.’

Lines 52-64: I would like you to better entertain the idea that coordination and collective motion as such may be largely a result of selection for grouping rather than selected for directly. Information acquisition may be an important factor as well but simple being and moving with others and thereby reducing predation risk and increasing foraging/energetic benefits may be the main driver for emerging collective motion.

We agree this view and that our previous manuscript did not reflect it correctly. We have now changed this statement (Lines 75-81):

‘Collective motion has evolved multiple times in fishes and is widely understood as a behavioral adaptation aimed at reducing the risk of predation². This behavioral adaptation is underpinned by the efficient acquisition of information from external cues through visual and lateral line sensory systems³⁻⁶. To date, we have a detailed understanding that coordinated movements in animal groups likely emerge from decision rules that individuals use to interact in groups^{e.g. 7,8}.’

Lines 66-68: Same as previous point. To what extent may we see coordinated movement in animal groups simply as a result of animals trying to be near others, and would we then really still expect the brain to play a major role in coordination? I think this would be very species and context specific.

We have rephrased to state that we think is important to evaluate brain anatomy as an important and understudied factor for the regulation of behaviors leading to the emergence of collective behavior (Lines 86-89):

‘The brain, as the central organ controlling locomotion, sensory systems and decision-making, can play an important role in the ability to coordinate movements and decisions made in the context of grouping.’

Lines 69-70: Here you are overlooking neurological work studying the onset of schooling in fish

Apologies, we have now restructured the introduction to combine our statement with examples of empirical studies explicitly testing the association between neuroanatomy and collective motion, with findings from those examples (Lines 99-108):

‘Despite its potential importance, empirical studies explicitly testing the association between neuroanatomy and collective motion are to date scarce. Brain regions associated with fish social behavior have also been identified in the few studies explicitly testing the link between neuroanatomy and schooling behavior. Lesion studies in goldfish (*Carassius auratus*) showed that individuals with ablated telencephalon exhibited reduced activity and association with conspecifics¹⁹. Also, a study on surface and cavefish populations of *Astyanax mexicanus* living in different light environments showed an underlying positive correlation between optic tectum size and schooling behavior differences between populations²⁰. These few previous studies highlight the potential role of neuroanatomy in schooling, as well as the need to account for environmental variation in analyses.’

Lines 92-93: See comment above, stick to one term for your selection experiment.

We have changed the text accordingly by using “polarization-selected” consistently.

Lines 97-102: Clarify here that the selection lines did not simply select for faster fish, which would also have higher polarisation due to its need to stay together at higher speed

We now provide a brief summary of combined factors that were found as the major causes driving the differences between polarization-selected and control summary (Lines 117-123):

‘In our selection lines, intrinsic schooling propensity was increased in female guppies by over 15% compared to controls in just three generations by selecting on individuals that exhibited higher polarization, the level of alignment between individuals moving together in a group^{19,22}. Previous assays in the polarization-selected females showed that differences in polarization

were caused by the combined effect of the likelihood to align with neighbors' movements, the strength of alignment to larger groups and individual swimming speed¹⁹.'

We likewise provide a more detailed account in the methods that selection for higher directional coordination was not only driven for selection for faster moving fish (Lines 550-557):

'Third generation polarization-selected females presented on average a 15% higher polarization, 26% higher median speed and 10% higher group cohesiveness (i.e. 10% shorter nearest neighbor distances) in comparison to control females¹⁹. Further tests in these lines showed that selection for higher directional coordination was not only driven for selection for faster moving fish. This is so because polarization-selected lines were still 5.7% more polarized after statistically controlling for speed differences, and the distance to conspecifics was an important factor driving differences in speed between polarization-selected and control lines¹⁹.'

Lines 112-114: Clarify here that those tests were with individual fish

We have changed the text accordingly (lines 126-131):

'For this, we performed a series of experiments and measurements to: i) evaluate collective motion patterns and predator inspection of individuals from polarization-selected and control female guppies when swimming in groups, ii) quantify brain region sizes with microcomputed tomography (microCT) of female guppies from polarization-selected and control lines, and iii) perform comprehensive tests of visual capabilities of these fish, spanning eye size, visual acuity and temporal resolution measurements of individual fish.'

Lines 126-137: Please state the sample size somewhere here in the main text as well as the body size of the fish, which is not provided anywhere as far as I can see.

We provided the total number of groups in the methods in the previous draft but we agree it is a good idea to state it at the beginning of results section and add information on body size. We have now added this information (Lines 149-153):

'In total, we obtained data for 83 polarization-selected groups and 81 control groups of approximately 6 months of age and of similar body size (polarization-selected: standard length mean [95% CIs] = 24.8 mm [24.0, 25.5]; control: standard length mean = 24.5 mm [23.8, 25.3]).'

Lines 145-152: It might be that the groups were already close to the minimum of inter-individual spacing, which Figure 1A suggests, with fish only being 2-3cm from one another. You could consider this possibility in the discussion, which is also interesting given the fact that the polarisation-selected fish manage to stay more polarised than control fish.

This is very likely, thanks for pointing this out! We have now added this possibility to the manuscript (Lines 162-167):

'Overall, nearest neighbor distance towards conspecifics in female showed a 10-fold increase in tests with a novel object and with a predator model in relation to OFT's (Supplementary Table 1, Fig. 1A). Such strong reduction in median nearest neighbor distance observed between

fish in the presence of these stimuli likely lead to minimum possible values of inter-individual distance in our experimental setup.'

Lines 156-158: What about the relationship between speed and polarisation? Please see my main point above with some critical thoughts and suggestions.

We have now added a section of the results describing the correlation between speed and polarization in our tests (Lines 162-167):

'Overall nearest neighbor distance towards conspecifics in female showed a 10-fold increase in tests with a novel object and with a predator model in relation to OFT's (Supplementary Table 1, Fig. 1A). Such strong reduction in median nearest neighbor distance observed between fish in the presence of these stimuli likely lead to minimum possible values of inter-individual distance in our experimental setup.'

Lines 176-182: For these results, and elsewhere, please also give an indication of the effect size in the text, e.g. "control lines spent on average 30% more time inspecting the fake predator than selection lines" [based on Fig 2D]. Also it is not clear here what is "selection: Ratio", is that the score of the selection lines divided by the score of the control lines? This should be clarified.

We apologize for the lack of clarity of our previous calculations and reporting of effect size in polarization-selected vs control lines comparisons. Ratio referred to the Incidence Risk Ratio, a parameter that provides information on effect size and that is commonly used for statistical models assessed via glmmTMB package. We have now clarified this (Lines 218-225):

'Specifically, we observed that polarization-selected females spent 21% less time inspecting the predator model and the mean duration of predator inspections were 18% shorter in polarization-selected females (GLMM_{time_inspecting}: Incidence Rate Ratio (IRR)_{polarization} = 0.79 (0.66-0.95), $t = -2.52$, $p = 0.011$; GLMM_{mean_inspection}: IRR_{polarization} = 0.82 (0.66-0.96), $t = -2.49$, $p = 0.013$, Fig. 2d-e, Supplementary Table 3): We observed a similar trend for total number of predator inspections performed with a 13% reduced frequency of inspections observed in polarization-selected females (GLMM_{inspections}: IRR_{polarization} = 0.87 (0.75-1.01), $t = -1.85$, $p = 0.064$; Fig. 2c, Supplementary Table 3).'

Lines 172-200: The order of the results presented here is not the same as in the figure; you start with figure 2d-e, then 2c, then 2a and 2b. Therefore either reorder the text or the figure panels so they are in the same order

We have now reordered the figure (Fig 3 in this manuscript) to match the order of results described in the manuscript.

Lines 210-216: Perhaps it would be good to clarify what areas you focused on and why. Is this more or less all (major) brain areas or did you leave out certain ones? And maybe also consider providing a short description of the key areas' function.

We think it is not ideal to provide brief statements on the function of each of the regions defined in our scans given that so little is known about the functions of regions in the guppy brain and across fishes. Yet, we have now clarified that for previous work in the lab, we created this guppy atlas with major areas that could be recognized in our scans (Lines 249-254):

‘We used microcomputed tomography (micro-CT) to reconstruct the brain anatomy of 13 polarization-selected females, and 15 control females and determine overall brain volume and the volumes of 11 major brain regions that could be safely identified in these brain scans covering the whole brain volume: olfactory bulbs, ventral telencephalon, dorsal telencephalon, thalamus, hypothalamus, nucleus glomerulus, torus semicircularis, optic tectum cup, central optic tectum, cerebellum, and medulla oblongata (see methods).’

Lines 210-214: Again please include some indication of effect size. From Figure 3 it seems the differences in size of the different regions are not very large?

We have now added effect sizes to all results including those on relative brain region volumes between polarization-selected and control lines (Lines 255-266):

‘Polarization-selected and control fish showed no differences in relative brain size (whole brain volume in relation to their body size; $LMM_{\text{relativebrain}}$: $t = -0.41$, $df = 23.29$, $p = 0.682$; Fig. 3a). However, analyses of relative brain region size (volume of each region in relation to the volume of the rest of the brain) indicated that the thalamus and optic tectum cups are approximately 7% and 4% larger in polarization-selected than control females respectively (LMM_{thalamus} : Odds Ratio (OR)_{polarization} = 0.929 (0.866-0.998); $t = 2.187$, $df = 25$, $p = 0.038$; LMM_{optectum} : Odds Ratio (OR)_{polarization} = 0.959 (0.924-0.994); $t = 2.409$, $df = 23.09$, $p = 0.024$; Fig. 3a), and the medulla oblongata is an 8% larger in control females (LMM_{medulla} : Odds Ratio (OR)_{polarization} = 1.08 (1.01-1.14); $t = -2.65$, $df = 23.91$, $p = 0.013$; Fig. 3a). All other eight brain regions measured presented no differences between polarization-selected and control females in relative region volumes (Fig. 3a-b, Supplementary Table 6).’

Figure 3: This figure can be improved by better clarifying the different regions of the brain in figure b, such as with a legend or by writing the abbreviations of the regions on the image, and reflecting the colors in the panels of figure A so that the reader can quickly link a plot to a certain region of the brain. Also, even with the right-most image in B it is still very hard to distinguish the TS, NG, and OTc so I suggest to replace it for a zoomed in image.

We have edited Figure 3 (Fig.4 in this manuscript) following suggestions from both reviewers. The right-most picture was an addition from previously prepared visualizations from an earlier study in which we developed the guppy brain atlas (Kotrschal et al. 2017). While we can still work with the brain atlas, we have no more access to the visualization software and unfortunately cannot edit this figure anymore or create a zoomed-in figure. This partially segmented picture was captured from a video we took during the segmentation process (that is also why some regions are not perfectly defined yet). As we agree that the partially segmented figure may seem confusing (and does not completely reflect our segmentation) we have removed it from the panel. We have additionally added text legends with abbreviations in each visible brain region with nomenclature used in the y axis of Panel A to facilitate readers to link each plot to the region of interest:

‘Fig 4. The effect of artificial selection for higher polarization in neuroanatomical allometric relationships. (A) The top left panel shows the allometric relationship between whole brain size volume and standard length of the fish (SL). Remaining panels show the relationship between each separate brain region with the rest of the brain ordered rostrally to caudally. Asterisks indicate brain regions with non-overlapping confidence intervals between polarization-selected females (pink; $n = 13$) and control females (blue; $n = 15$) in two consistent statistical analyses (Supplementary Tables 6-7). **(B)** Reconstructed brain regions from microCT - scanned guppy brains. A dorsal (left) and lateral (right) view of a guppy brain with the major brain regions color coded: olfactory bulbs (OB; dark blue), dorsal telencephalon (DT; red), ventral telencephalon (VT; light green), optic tectum (OT; yellow), hypothalamus (Hyp; turquoise), thalamus (Th; purple), cerebellum (Cb; brown), medulla oblongata (MO; dark green). Not visible but measured brain regions include: torus semicircularis, nucleus glomerulus and optic tectum core.’

Reference cited:

Kotrschal, Alexander, et al. "Evolution of brain region volumes during artificial selection for relative brain size." *Evolution* 71.12 (2017): 2942-2951.

Lines 226-235: It is not immediately obvious to me how these additional analyses add to the results given above while they do seem to add a lot of statistical tests. Please better clarify how this result of a negative correlation between the two areas extends beyond finding they were different in the selection lines as well as how maybe with your bayesian approach you could account for type I error by running a large number of correlations.

We think it is important to evaluate allometric relationships between brain regions that showed changes following artificial selection and add important findings to debates on brain evolution in this regard. All results reported from these correlations are based on posterior probabilities from a Bayesian multilevel statistical method that modelled

residual correlations among regions and therefore no multiple statistical tests are performed with this approach.

Line 246: Clarify if these 112 individuals were also used in the schooling experiments and if the experiment was indeed performed after the group trials.

We did not reuse fish across collective motion and visual performance experiments or across these and brain morphology measurements. We have now added this information in the methods section (Lines 560-565):

‘All fish were removed from their parental tanks after birth, separated by sex at the first onset of sexual maturation, and afterwards transferred to single-sex groups of eight individuals in seven-liter tanks. We kept all fish used for anti-predator response experiments, visual capacity tests and brain morphology measurements in these groups throughout their life span. Fish were not reused for experiments across these three categories.’

Lines 253-266: This paragraph is missing a straight-to-the point explanation of the test and what it measures. Also a (supplementary) figure of the setup would be helpful.

The optomotor response test is a widely used method to measure visual acuity in fishes (also in guppies) and we have successfully implemented it in the lab using this same methodology to measure several components of visual perception of guppies. We provided this argumentation in the manuscript and direct the reader to our previous papers/manuscripts where extended details of the methods we used in the lab are provided including setup figures. However, we have now provided further information on the expectations of this test following the reviewer’s next comment (lines 320-326):

‘If polarization-selected and control groups differed in their ability to resolve spatial ability, differences in their optomotor response towards these rotating stimuli at the lower end of the guppy visual acuity should be expected. However, we found no difference in their average optomotor response combining data from all stimuli ($LMM_{\text{acuity}}: \text{Estimate}_{\text{selection}} = 0.00 [-0.08, 0.009]$, $t = 0.11$, $df = 12.88$, $p = 0.913$; Supplementary Table S9a), or in analyses independently evaluating specific optomotor response for any of the six stimuli presented (Fig. 5B; Supplementary Table 9c).’

Lines 274-281: The same. What are you expecting here from the test? You don't explain what outcomes the test actually provides and how. For example, line 281 states the fish followed the direction of the rotating stimulus, but it is not clear to me

We have provided extended details on methods and expectations form this test (Lines 335-342):

‘Using the same experimental apparatus used to evaluate visual acuity, we video recorded female guppies from our selection and control lines and exposed them to a single-width rotating stimulus that moved in multiple directions and at different speeds (see methods). We next used automated tracking to obtain orientation and speed of the fish for each frame and to quantify their direction and speed in relation to the stimuli presented at each time point. If polarization-selected and control groups differed in their ability to track movement, differences in the time following the correct direction of the stimuli and the deviation of their speed in relation to the stimulus speed should be expected.’

In addition, we understand the problem to interpret this statement might have been originated because Supplementary Figure 2 wrongly depicted speed deviation duplicated in top and bottom panels, instead of the proportion of time following stimulus in panel A and speed deviation in panel B as originally intended and described in the previous manuscript. We have now edited Supplementary Figure 2 to include the proportion if time data:

Fig S2. Temporal resolution of female guppies artificially selected for higher polarization. Boxplots and density plots of the deviation of fish swimming speed (**A**), and the proportion of time that fish followed the direction of the stimulus (**B**) in relation to four different rotating stimuli presented at different speeds that rotated clockwise and anti-clockwise to polarization-selected (pink; $n = 58$) and control females (blue; $n = 56$). In the boxplots, horizontal lines indicate medians, boxes indicate the interquartile range, and whiskers indicate all points within 1.5 times the interquartile range. No

significant differences were observed for any comparison between control and polarization-selected fish (see Tables S9-S10).

Lines 307-309: You cannot make this statement. Your inspection analyses focused on individuals only so you cannot say that fish spent less time inspecting individually VS socially, implied here by stating 'selected groups rely more on neighbour information'. It would have been good to investigate the inspection tendency of all individuals over time, something that is possible with the present data, and thereby actually be able to make a statement as given here.

We agree and have now changed this statement in the text (Lines 367-372):

‘Our work demonstrates that selection for schooling behavior in female guppies has important implications for how this species behaves in the presence of potential threat. Analyses of motion patterns in these fish show that polarization-selected groups maintain higher polarization and speed when exposed to a potential threat. In addition, our analyses indicate that individuals from polarization-selected groups spend less time inspecting the threat than individuals from control lines.’

Lines 309-312: To me it is a bit too far to go from not finding any differences in these abstract tests with moving lines to the interpretation that thus polarisation-selected fish are not better able at distinguishing threats at longer distances or acquire visual information of neighbours. Stick closer to what the tests actually measure and discuss the results as such.

We respectfully disagree with the reviewer’s interpretation of the biological relevance of the tests performed on visual performance. Our methods on optomotor response are in line with methods used to assess visual performance across multiple species of fishes and other organisms and can provide important information on the ability of polarization-selected and control lines to acquire sensory input in biologically-relevant situations. We have however changed the statement to reflect what our tests measured more exactly (Lines 372-376):

‘We further studied visual capacities in these fish and find no differences between polarization-selected and control individuals in visual acuity, temporal resolution or eye size, suggesting that effective processing of visual information in the brain is key for the observed differences in their ability to synchronize their swimming with close neighbors in multiple contexts.’

Line 314: Again, here you are talking about selection for higher schooling propensity rather than polarisation

We have changed the text accordingly by using “polarization-selected” consistently.

Lines 361-365: This is not clear. What do you mean with 'activity' and how do you mean activity is primarily associated with the exchange of directional information? And why do you interpret finding a difference in activity as suggesting there are differences in the ability to detect neighbour movements or threats? I don't follow these interpretations.

We have removed this sentence and clarify our interpretation on the role of speed in collective motion across contexts, which now we believe is more complete thanks to the reviewer’s suggestion on further analyses between these two variables in our experiment.

Lines 404-420: As some of the main functions of the thalamus are to relay sensory information, to me it seems more logical to also interpret the enlarged thalamus in polarisation-selected females in the same direction as the enlarged optic tectum cup; that these fish are better able at processing (visual) information (from others) and are thereby able to better align with others while on the move. Currently this section primarily focuses on potential for its role in predator inspection behaviour and aggression, which is less likely in my opinion. Please consider and discuss further.

We agree and now focus more on the potential role of the thalamus in relaying processed sensory information than on its potential role regulating fear response (Lines 469-494):

‘The primary receiving region of processed inputs from the tectum, the thalamus, was also found to be enlarged in polarization-selected females. To date, the level of homology between the mammalian and teleost thalamic regions is under strong debate^{55,56}. In mammals, the thalamus plays critical roles in the modification, filtering and distribution of sensory and mechanosensory information into decision-making regions of the brain^{56,57}. Recent findings in zebrafish suggest that these tasks are performed mainly by the preglomerular complex, whose developmental origin might be outside the diencephalic area that contains the thalamus in mammals⁵⁸. The neuroanatomical atlas we created to analyze relative region size does not allow us to assess whether this particular region is part of the thalamus area which has been enlarged in polarization-selected females. However, our results of strong gene expression changes in the telencephalon of polarization-selected and control females in response to decision-making across social contexts⁵⁹, are in agreement with a higher efficiency in the relay of processed information towards the telencephalon, potentially facilitated by an enlarged thalamus. Studies across teleosts and mammals do agree on that the thalamus have an important role in the development of glutamatergic and GABAergic relay nuclei, with crucial roles in neurotransmission and inhibition. Interestingly, functional characterization of genomic and transcriptomic analyses performed in these selection lines identified a crucial role of glutamatergic synaptic function in the evolution of schooling⁵⁹. Taken together, our results suggest that changes in the development of glutamatergic relay nuclei driven by a larger thalamus might affect the ability of individuals to synchronize their movements in a shoal. In parallel, and while not directly addressed in this study, the regulatory role of the thalamus in aggressiveness is concordant with common expectations of lower aggression levels in group-living species (reviewed in ⁶⁰). As such, further quantifications of the anatomical characteristics of the thalamus in relation to aggression levels within and across species might be a promising avenue for future research.’

Lines 431-434: The argument given here is not really correct. That fish in groups startle less than those alone is to do with direct anti-predator benefits of grouping - at the proximate level - while your observation that polarisation-selected fish show less predator inspection is more at the ultimate level and not much can be said about changes at the proximate level due to trials with individual fish lacking in the present study.

We agree with the reviewer’s interpretation and we have now removed references to the potential role of the medulla in startle behavior in the discussion.

Lines 458-464: These concluding sentences correctly reflect the main findings and interpretation of the study.

We agree, and we think that edits in the discussion with less focus on social information processing in response to predation now reflect these interpretations better.

Lines 477-479: Also clarify here how the polarization-selected females differed in speed from control lines.

We have clarified this point in the methods too as suggested (Lines 551-558):

‘Third generation polarization-selected females presented on average a 15% higher polarization, 26% higher median speed and 10% higher group cohesiveness (i.e. 10% shorter nearest neighbor distances) in comparison to control females¹⁹. Further tests in these lines

showed that selection for higher directional coordination was not only driven for selection for faster moving fish. This is so because polarization-selected lines were still 5.7% more polarized after statistically controlling for speed differences, and the distance to conspecifics was an important factor driving differences in speed between polarization-selected and control lines
21.

Lines 493-494: You used super shallow water depth of only 3cm, please see my main comment about this.

As we discuss above, the shallow water depth in our experimental setup elicited no indications of abnormal behavior in fish. Furthermore, in our experience performing field collections of guppies in natural habitats, it is common to find streams with similar water depth in which guppies inhabit. We additionally performed a non-exhaustive search for work from other authors and found examples of similar water depths in guppy experimental setups: Evans and Magurran (2000): schooling behavior - 2.5 cm; Ghalambor et al. (2004): swimming performance and escape response - 3 cm; Feyten et al. (2021): decision-making in relation to predation - 3 cm.

Overall, we believe this is not an issue of concern.

Lines 499-500: Provide details about the length and height of the model and how it was kept into place.

We have now added details in relation to this (Lines 582-588):

‘We used a blue coffee mug as a novel object and a fishing lure (18 x 3 cm) custom-painted to resemble the pike cichlid *Crenicichla frenata*, a natural predator on the guppy, as the predator model. These objects have been previously used to successfully reproduce natural behaviors of the guppy in response to a novel object and a predation threat⁴⁷. To facilitate automated data collection, the position and orientation of the predator model was kept constant by using magnets in the bottom of the fishing lure and in the central position of the experimental arena.’

Lines 497-501: Was the orientation of the mug and predator model always presented in the same orientation or was it randomized? Without it, how can you account for arbitrary spatial preferences in the tank in your analyses, such as those related to front-back position?

In short, we decided to maintain same orientation of stimuli to facilitate automated tracking data collection and we cannot account for arbitrary spatial preferences. However, we do not believe this is a major issue in our experimental setup. Given the large number of trials performed over the years in this setup to generate the selection lines, we have put a strong emphasis on maintaining the same level of illumination across the whole arena which is a key factor in obtaining reliable data from idTracker. Likewise, the setup was built in an isolated room of the lab where disturbances were minimal. As such, we understand that spatial locations should be mostly affected by the presence of the stimuli in the experimental arena. We have added this information to the text to clarify that arbitrary spatial preferences might have played a role in our results (Lines 586-591):

‘To facilitate automated data collection, the position and orientation of the predator model was kept constant by using magnets in the bottom of the fishing lure and in the central position of the experimental arena. We provided external illumination to avoid shadowed areas in the circular arena. Disturbances were minimized by performing all assays in an isolated room of the laboratory. Under these conditions, the presence of stimuli should be a major driver of the

spatial locations of fish in the experimental arena. However, we are unable to quantify the effect of arbitrary spatial preferences in our results.’

Lines 510-511: Please just state the three variables you measured; I suggest to not suddenly start talking about these possible 'axes' of collective motion and polarisation does not necessarily have to be a 'sociability axis'

We have now removed the reference to guppy collective motion axes and refer to polarization, speed and nearest neighbor distance throughout the manuscript.

Lines 514-516: In this case your measure of polarisation could be biased as you are losing information on alignment when fish are further away from one another. This is especially critical as polarisation is the main measure of your study and used for creating the selection lines. You need to provide statistics to show the proportion of time groups were beyond the threshold and how this differed between the lines, how your subsetted polarisation measure is related to polarisation without any subsetting, and an explanation why this threshold was used. This is an important point that needs addressing.

We understand the reviewer concerns about the validity of the data to compare polarization-selected and control collective motion characteristics. We have now clarified the steps taken to extract the two different data sets used for collective motion in analyses and heatmaps, as well as the steps we took to ensure that our data extraction did not introduce biases in the comparisons between polarization-selected and control groups (Lines 598-622):

‘We tracked the movement of fish groups in the collected video recordings using idTracker⁶⁹ and used fine-grained tracking data to calculate speed, polarization and nearest neighbor distance in Matlab 2020 following methods established in (70). For speed, we obtained the median speed across all group members by calculating the first derivatives of the x and y time series, then smoothed using a third-order Savitzky–Golay filter. For nearest neighbor distance, we obtained the median distance to the nearest neighbor for every fish across all frames. For group polarization, we calculated the median global alignment, which indicates the angular alignment of all fish in the arena. Calculations of median global alignment in each frame were only calculated if six out of the eight members of the group presented tracks following the optimization of our tracking protocol in the setup in (24). No differences between completeness of tracks were observed between polarization-selected and control groups for any of the treatments in our experiment (Supplementary Figure 3). For all measurements trials with less than 70% complete tracks were disregarded for further analyses. We additionally used R (v4.1.3), RStudio (v2022.07.0) and the tidyverse package^{71–73} to generate heatmaps with average values of group polarization and group centroid within 20 x 20 mm grid cells for each frame (Fig. 1B). To avoid biases in group centroid measurements potentially resulting from single individuals or small subgroups at large distances, we limited the analyses to frames in which at least six individuals formed a connected group, with an interindividual distance of 10cm counting as a connection. Grid cells that did not contain values for a minimum of 8 groups per treatment were disregarded. No differences between the proportion of frames used were observed between polarization-selected and control groups for any of the treatments (Supplementary Figure 3). To evenly compare motion patterns when presented with a novel object and a predator model to those obtained during the open field assays, we limited our analysis of the open field assay data to the initial six minutes of the recording.’

We additionally now provide data on no differences observed in the completeness of tracks and proportions of frames used between polarization-selected and control groups across treatments:

Fig S3. Tracking data reliability between groups. We found no differences between polarization-selected groups (pink) and control groups (blue) in (A) the total proportion of frames tracked by idTracker in open field, novel object and predator model assays, or in (B) the data used to generate heatmaps that only considered frames in which at least six individuals formed a connected group, with an interindividual distance of 10cm counting as a connection.'

Lines 517-518: How often did this happen? Again, it is important you provide the proportion of time you have correct tracking data for and indicate if this was not different between the lines and the different tests. And why did you use a threshold of 16 frames?

We refer to the previous comment that also answers the first part of this question. As for the 16 frames threshold, it was never used as a filter for data reliability for either data extraction in Matlab or heatmap visualizations in R. This mistake was caused by a miscommunication between authors about the application of a Savitzky-Golay filter that is used during data extraction from tracking only to smooth median speed data across series and not for measurements of polarization or attraction. For the correct application of this filtering procedure in R, a width of 13 frames is needed (instead of the 16 frames wrongly stated in our previous version).

Lines 518-520: It is not clear how this is different from that states on 514-516. Do you mean polarisation without subsetting?

We think edits presented above in relation to data extraction clarify also that we performed two separate data analyses from tracking data in relation to polarization:

- 1) to calculate polarization for each group measured at each frame across treatments (performed in Matlab).**
- 2) to visualize the association between group polarization and spatial position in the experimental arena in grids of 20x20 mm across treatments (performed in R).**

Lines 546-553: How much insight can we properly get into inspection behaviour by just looking at the behaviour of one individual in a group of 5? We lose any information about the social aspect of inspection behaviour, which should actually be very interesting in the context of the present work. Is it just that polarised-selected individuals inspect the predator less irrespective of being with others or not or that the evolved differences in social behaviour affect individuals' inspection behaviour. From the present analyses this cannot be disentangled.

Following this and previous comments from the reviewer, we proceeded to remove references to social information acquisition based on predator inspection data.

Lines 592-596: More (background) information should be provided here (or in the main text) about the brain regions. Why did you focus on these 11? Is the reason that together they comprise the majority of the fishes' brain? See also my more general comment.

The neuroanatomical methods were a replication of previous work in the lab and we directed the reader to a previous article with extended details on this information. We have however added brief information on this regard as we think the reviewer is correct in that some background information would be useful here (Lines 694-704):

‘This protocol allowed us to obtain measurements of whole brain size volume and brain region volume in 11 major brain regions in the guppy brain: olfactory bulbs, ventral telencephalon, dorsal telencephalon, thalamus, hypothalamus, nucleus glomerulus, torus semicircularis, optic tectum cup, central optic tectum, cerebellum, and medulla oblongata (Fig. 4B). We chose all regions that we could safely identify in the brain scans based on an adult swordtail brain atlas⁸⁰ and our own knowledge in fish neuroanatomy. Extended details on guppy brain region reconstruction from digital images can be found in ⁽⁵³⁾.’

Lines 626-634: Still a lot of key information is missing here, making it hard to properly understand what was done. So the projection was 360 degrees on all the walls thus surrounding the fish? How did you achieve this? What do you mean with "rotating bands", that the bands move along the walls in circles? And is the stimulus only of a couple bands that thus move around or what do the fish actually see? And foremost, why would the fish respond

to one type of width stimulus and not the other, how is that linked to acuity? What do you expect the fish to not see when presenting moving bands?

We understand that the reviewer, as well as other readers, might be unfamiliar with previous work on the use of the optomotor response for quantification of visual performance, and we apologize if we did not sufficiently explain it. Our work is not novel in this respect and we reproduced in this study a widely used test in fishes (and many more taxa) to assess visual performance. Although we respectfully suggest that it is unnecessary to provide a detailed characterization on the optomotor response test used here, we have however briefly clarified our methods in the text and provided references to a review that extensively discuss the use of this test to assess visual performance (Lines 732-745):

‘We evaluated the ability to perceive detail (visual acuity) in 9-12 months old female guppies (polarization-selected: n = 59, standard length [95% CIs] = 270 mm [266, 274], control: n = 57, standard length [95% CIs] = 271mm [268, 275]). For this, we assessed their optomotor response, an innate orient behavior induced by whole-field visual stimulation⁸⁴, a widely used method to assess visual acuity across tax (reviewed in ⁸⁵). Extended methods and the optimization procedure for the stimuli used here can be found in (²⁸). Briefly, we projected a video recording with rotating vertical black and white bands of six different widths (stimuli) on the walls of a white ring-shaped arena of 25/50 cm of inner/outer diameter. Previous optimization of the methods found that the use of these stimuli allowed us to evaluate the optomotor response at the lower end of the species’ acuity²⁸. We placed individual fish in between the inner wall of the arena and a transparent ring of 40 cm diameter. After a two-minute acclimation period, we recorded their response towards six different rotating stimuli and the static images of these stimuli using a Sony Cam HDR-DR11E recorder. Each stimulus was presented for one minute in random order.’

Line 653: You mean the bands were projected 360 degrees on the walls?

We used one of the projections previously used in the visual acuity tests following same methodology. We have clarified this fact and that it was likewise projected in the walls (Lines 758-763):

‘To evaluate temporal resolution, we placed fish in a white arena (50 cm diameter, 4 cm water depth) and exposed them to a projection of black and white bands of 3.5 cm width on the walls rotating clockwise and counterclockwise at four different angular speeds (14.4, 25, 36 and 45 degrees/sec). We chose this bandwidth to focus on the assessment of fish response towards direction and speed changes, as this stimulus showed a maximal optomotor response in prior visual acuity tests²⁸.’

Reviewers' Comments:

Reviewer #1:

Remarks to the Author:

The authors have responded satisfactorily to my concerns on the original version of this MS and I am pleased to recommend this exciting work for publication.

Reviewer #2:

Remarks to the Author:

The authors have done a great job tackling my and the other reviewer's comments and took a lot of care in formulating their replies. The authors carefully assessed each of my points and it is good to see the majority of them integrated to help improve the manuscript. But it is also fair that the authors respectfully disagreed on some points and provided a thorough explanation of their reasoning. As I also already highlighted in my original review, I think this paper is a very impressive and elaborate piece of work with a strong empirical approach and thought-provoking results. I do believe this paper is now worthy of being published in *nature Communications*. I only have one general comment left.

One of my main original comments was that there was too much focus on the "anti-predator behaviour" aspect of the study while its responses in a (fake) predator context are only one part of the study and the study is not really designed to disentangle different underlying mechanisms for differences in polarisation and inspection behaviour in this context, especially as you only looked at the later in a social group and with a randomly selected individual. You generally followed this point and have revised your manuscript accordingly, but I still found it falls short at a couple of places which may result in readers expecting your paper to be providing detailed insights into anti-predator behaviour, while its focus lies more on the across-context consistency in polarization and associated changes in selected brain areas. Hence I think you could still try and describe the focus of your study and the interpretation of the results more accurately, especially in the abstract on lines 41-44 and 50-52, and in the introduction on lines 124-126. Finally, be careful with the sentence in the discussion on lines 386-388 "how the association between brain morphology and anti-predator behavior might drive the evolution of collective behaviour." as this implies a directional effect while it could also be the other way around or the observed differences in behaviour in the predation context being a by-effect, such as because of different information processing.

PS: For future revisions, it would be helpful for the reviewers if you better highlighted the edited text in the revised manuscript, such as by using a different colour or underlining, as the dark blue you used now is hardly discernable from black.

REVIEWERS' COMMENTS

Reviewer #1 (Remarks to the Author):

The authors have responded satisfactorily to my concerns on the original version of this MS and I am pleased to recommend this exciting work for publication.

Reviewer #2 (Remarks to the Author):

The authors have done a great job tackling my and the other reviewer's comments and took a lot of care in formulating their replies. The authors carefully assessed each of my points and it is good to see the majority of them integrated to help improve the manuscript. But it is also fair that the authors respectfully disagreed on some points and provided a thorough explanation of their reasoning. As I also already highlighted in my original review, I think this paper is a very impressive and elaborate piece of work with a strong empirical approach and thought-provoking results. I do believe this paper is now worthy of being published in *Nature Communications*. I only have one general comment left.

One of my main original comments was that there was too much focus on the "anti-predator behaviour" aspect of the study while its responses in a (fake) predator context are only one part of the study and the study is not really designed to disentangle different underlying mechanisms for differences in polarisation and inspection behaviour in this context, especially as you only looked at the later in a social group and with a randomly selected individual. You generally followed this point and have revised your manuscript accordingly, but I still found it falls short at a couple of places which may result in readers expecting your paper to be providing detailed insights into anti-predator behaviour, while its focus lies more on the across-context consistency in polarization and associated changes in selected brain areas. Hence I think you could still try and describe the focus of your study and the interpretation of the results more accurately, especially in the abstract on lines 41-44 and 50-52, and in the introduction on lines 124-126. Finally, be careful with the sentence in the discussion on lines 386-388 "how the association between brain morphology and anti-predator behavior might drive the evolution of collective behaviour." as this implies a directional effect while it could also be the other way around or the observed differences in behaviour in the predation context being a by-effect, such as because of different information processing.

PS: For future revisions, it would be helpful for the reviewers if you better highlighted the edited text in the revised manuscript, such as by using a different colour or underlining, as the dark blue you used now is hardly discernable from black.

We are thankful to both reviewers for their work and for valuing our work in revising the manuscript. Following the reviewer recommendations, we have revised these sentences in the main text (underlined text):

Abstract:

“One of the most spectacular displays of social behavior is the synchronized movements that many animal groups perform to travel, forage and escape from predators. However, elucidating the neural mechanisms underlying the evolution of collective behaviors, as well as their fitness effects, remains challenging. Here, we study collective motion patterns with and without predation threat and predator inspection behavior in guppies experimentally selected for divergence in polarization, an important ecological driver of coordinated movement in fish. We find that groups from artificially selected lines remain more polarized

than control groups in the presence of a threat. Neuroanatomical measurements of polarization-selected individuals indicate changes in brain regions previously suggested to be important regulators of perception, fear and attention, and motor response. Additional visual acuity and temporal resolution tests performed in polarization-selected and control individuals indicate that observed differences in predator inspection and schooling behavior should not be attributable to changes in visual perception, but rather are more likely the result of the more efficient relay of sensory input in the brain of polarization-selected fish. Our findings highlight that brain morphology may play a fundamental role in the evolution of coordinated movement and anti-predator behavior.”

Line 126-129: “Following directional selection for polarization, we used these lines to investigate collective motion characteristics with and without predation threat, as well as to study the association between increased schooling propensity and changes in brain anatomy and visual performance.”

Line 383-385: “Below, we discuss the implications of these discoveries for potential associations between brain morphology, anti-predator behavior and the evolution of collective behavior.”

We apologize for the choice of color of the revised text in our previous response and take this recommendation for future revisions.